

# Quench dynamics of the Ising field theory in a magnetic field

**Kristóf Hódsági[1], Márton Kormos[1,2] and Gábor Takács[1,2]**

**1** Department of Theoretical Physics, Budapest University of Technology and Economics,
1111 Budapest, Budafoki út 8, Hungary
**2** BME "Momentum" Statistical Field Theory Research Group,
1111 Budapest, Budafoki út 8, Hungary

## Abstract

We numerically simulate the time evolution of the Ising field theory after quenches starting from the $E_8$ integrable model using the Truncated Conformal Space Approach. The results are compared with two different analytic predictions based on form factor expansions in the pre-quench and post-quench basis, respectively. Our results clarify the domain of validity of these expansions and suggest directions for further improvement. We show for quenches in the $E_8$ model that the initial state is not of the integrable pair state form. We also construct quench overlap functions and show that their high-energy asymptotics are markedly different from those constructed before in the sinh/sine–Gordon theory, and argue that this is related to properties of the ultraviolet fixed point.


doi:10.21468/SciPostPhys.5.3.027

# 1  Introduction

Recent years witnessed significant advances in understanding the out of equilibrium dynamics of isolated quantum many-body systems [1–3]. On the experimental side, it has become possible to routinely engineer and manipulate isolated quantum systems using cold atomic gases [4–12]. This opened the way to the experimental study of coherent time evolution which induced immense theoretical investigations.

One of the simplest non-equilibrium situation corresponds to a sudden change of some parameters of a system prepared in an equilibrium state, typically in its ground state. This is called a quantum quench and has become a paradigm of non-equilibrium dynamics [13, 14].

For a long time, the focus of the theoretical investigations was the description of the late time asymptotic steady state. A key result of these studies is that there is a dichotomy between integrable and non-integrable systems: while generic systems locally thermalize after a quantum quench, integrable systems approach a steady state which can be described in terms of a Generalized Gibbs Ensemble (GGE) constructed from the local conserved quantities [15]. Specifying the complete set of conserved charges in interacting integrable models proved to be a non-trivial problem [16–22].

It is a natural question what happens in systems that are close to being integrable. In classical mechanics, this is the subject of the Kolmogorov–Arnold–Moser (KAM) theory that explains, for example, the stability of our solar system on a large time scale.The extension of the KAM theory to quantum systems is an open problem [23].

Beyond the steady state, the description of the actual time evolution is also of interest. An important question is whether universal features of the non-equilibrium dynamics can be identified. For example, a two-step relaxation process, the so-called prethermalization scenario was proposed to capture the time evolution in weakly non-integrable systems [7,24–28]. Theoretical description of the out of equilibrium time evolution is much more difficult and less understood than the characterization of the steady state. Analytical results have been obtained in systems that can be mapped to free particles [29–43] and in conformal field theory [13,14]. For small quenches, the semiclassical approach can yield analytic results [44–48]. In quantum

field theories, approaches based on form factor expansions [49–53] have been developed in the small quench limit.

Quantum field theories provide an effective description of quantum systems near their quantum critical point and capture universal behavior [54]. Small quenches in the vicinity of the critical point are expected to be described by quantum field theories which may thus capture universal physics even out of equilibrium. Apart from this, quenches in quantum field theories are interesting in their own right and are relevant to high energy physics and cosmology [55].

Even though important aspects of field theory quenches have been understood [32, 56–64], the field of quenches in quantum field theories is less developed compared to lattice systems and spin chains. One reason for this is the lack of effective numerical methods in continuum systems. Recently, a Hamiltonian truncation method called Truncated Space Approach (TSA) was developed and applied to quenches in the Ising field theory [65]. This is a non-perturbative method that is suitable for studying perturbed conformal or free field theories, independently from their integrability. The essence of the method is the exact calculation of the matrix elements of the finite volume Hamiltonian in the conformal or free basis, followed by the exact diagonalisation of the finite Hamiltonian matrix obtained by cutting off the spectrum at some energy value. The TSA method has been applied successfully to the study of the spectrum of perturbed minimal conformal field theories [66–69], the sine–Gordon model [70], Landau–Ginzburg models [71–74], Wess–Zumino models [75–77]. It has also been used to study integrability breaking in the 1D Bose gas [23].

Although there have been some other approaches to quantum quenches incorporating Hamiltonian truncation, either combined with Bethe Ansatz techniques in one-dimensional Bose gases [23] or as a part of a chain array matrix product state algorithm in two dimensions [78], the work [65] was the first to simulate the time evolution solely based on the TSA approach. Very recently, the method was used to study multipoint correlation functions after quenches in the sine–Gordon model [79].

In this work we take the next step in the Ising field theory and apply the Truncated Conformal Space Approach (TCSA) in a different regime of the Ising field theory, namely, we study quenches in and near Zamolodchikov's $E_8$ integrable field theory [80]. The name of the model reflects the fact that the masses and the scattering amplitudes of the eight quasiparticle excitations can be described in terms of the $E_8$ exceptional Lie group. This massive field theory emerges as the scaling limit of the quantum Ising spin chain in a small longitudinal magnetic field when the transverse field is tuned to the quantum critical point. This system has been realized experimentally in $CoNb_2O_6$ [81]. The quenches we study here correspond to sudden changes of the transverse and longitudinal magnetic field in the spin chain.

We investigate the time evolution of various observables in quenches to both integrable and non-integrable Hamiltonians. We compare our numerical results with two different analytic descriptions. Both of them are form factor expansions but they differ in important aspects.

Delfino's approach [51, 52] is a perturbative one that exploits the integrability of the *pre-quench* system but does not require integrability of the post-quench Hamiltonian. The perturbative expansion has been worked out only to first order so far and this is what we compare with our numerical results. The first order approximation is expected to be valid for long times when the quench amplitude approaches zero. Since this is the only analytic approach to quenches to non-integrable systems, it is paramount to investigate the precise domain of validity of the first order result and the nature of its deviations from numerical results for larger quenches. These can carry precious information about the potential of higher perturbative orders and suggest directions for further developments.

The other approach relies on the integrability of the *post-quench* system. In contrast to the perturbative approach, it is not a first principles method as it needs input about the initial state.

In its most powerful version developed by Schuricht and Essler [49, 50, 53], a resummation of the infinite form factor series is carried out by which it becomes a large time expansion valid up to infinite time. In particular, it can capture the temporal relaxation of expectation values. This approach is expected to be valid for appropriately small quenches but it is not perturbative as its small parameter is the post-quench particle density rather than the change in the coupling. Both methods can be applied to small quenches between integrable systems, e.g. to a parameter quench in an integrable model, which allows for a comparison of the two methods using the numerical TSA results.

Let us point out that the majority of the analytic results in integrable models both for the asymptotic state in terms of a GGE and for the time evolution have been obtained for a special class of initial states which can be viewed as a superposition of parity invariant states of quasiparticle pairs having opposite momenta. A more rigorous criterion for such "integrable" initial states for Bethe Ansatz solvable models was given in Ref. [82]. It was shown in Ref. [60] that in relativistic field theories these states necessarily have an exponential "squeezed state" form. However, it is an interesting question whether quenches that preserve integrability of the Hamiltonian can lead to non-integrable initial states, and in fact it was argued in Ref. [52] that this is indeed generally the case.

The paper is organized as follows. After introducing the model and the TCSA method in Sec. 2, we start our investigations with quenches preserving integrability in Sec. 3, where we compare the numerical data with the predictions of the two analytic approaches. In Sec. 4 we turn to quenches that break integrability both in the ferromagnetic and paramagnetic phases. In Sec. 5 we analyze the overlaps of the initial state with the multiparticle states of the post-quench Hamiltonian. Apart from checking the perturbative predictions, we also look for exotic multiparticle states and study the energy dependence of these overlaps. We give our conclusions and outlook in Sec. 6. Important facts about the $E_8$ model as well as details of the TSA method can be found in the appendices.

## 2 Model and numerical method

We study the non-equilibrium dynamics of the scaling Ising field theory which is a perturbation of the $c = 1/2$ conformal field theory by its two relevant fields $\epsilon$ and $\sigma$, described by the formal action

$$\mathcal{A} = \mathcal{A}_{\text{CFT}, \, c=1/2} - h \int \sigma(x) d^2x - \frac{M}{2\pi} \int \epsilon(x) d^2x \,. \tag{2.1}$$

This field theory arises as the continuum limit of the quantum Ising spin chain in the vicinity of its quantum critical point. The conformal field theory describes the critical point, the $\epsilon(x)$ operator corresponds to the transverse magnetization and $\sigma(x)$ is the continuum limit of the longitudinal magnetization operator.

For the case $h = 0$ the model describes a free Majorana fermion of mass $M$, while the parameter $h$ corresponds to a longitudinal magnetic field coupled to the magnetisation $\sigma$. The choice of parameters $M = 0$ and $h \neq 0$ results in the famous $E_8$ model with 8 stable massive particles [80]. In this case the mass $m_1$ of the lightest particle is related to the magnetic field $h$ by [83]

$$m_1 = (4.40490857\dots)|h|^{8/15}. \tag{2.2}$$

In this paper we study quenches starting from the $E_8$ axis where the post-quench system is governed by the action

$$\mathcal{A} = \mathcal{A}_{CFT, \, c=1/2} - h_i \int \sigma(x) d^2x + \lambda \int \Psi(x) d^2x \,, \tag{2.3}$$

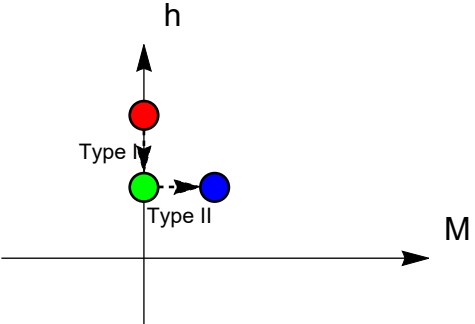

Figure 2.1: Illustration of quenches considered in this work in the $M - h$ parameter space. The lower (green) dot on the axis represents the post-quench Hamiltonian for Type I quenches and the pre-quench Hamiltonian for Type II quenches which were performed in both (negative and positive $M$) directions.

where $\Psi(x)$ is the quenching operator, and the initial state is the ground state of the theory with $\lambda = 0$.

We shall consider two classes of quenches which are illustrated in Fig. 2.1. The first class (Type I) consists of quenches performed along the $E_8$ axis $M = 0$ which correspond to changing the magnetic field from an initial value $h_i$ to $h_f$, so $\Psi(x) = \sigma(x)$ and $\lambda = h_i - h_f$. The magnitude of these quenches can be parameterised using the dimensionless combination

$$\xi \equiv \frac{\lambda}{h_f} = \frac{h_i - h_f}{h_f} . \tag{2.4}$$

The dimensionful $h_f$ is not a good parameter: it can be rescaled by a choice of units for energies and length scales. Using (2.2) we fix $h = h_f$ such that the mass gap $m_1$ of the post-quench Hamiltonian is set to unity, $m_1 = 1$.

The second class (Type II) of quenches corresponds to switching on a non-zero $M$ while keeping the magnetic field $h$ fixed, so $\Psi(x) = \epsilon(x)$ and $\lambda = -M/(2\pi)$. The resulting action is not integrable so these quenches break integrability.

The quench protocol is implemented in the Truncated Conformal Space Approach (TCSA). (Details on TCSA are given in Appendix B.1.) The method is based on the exact calculation of the matrix of the finite volume Hamiltonian that governs the time evolution exploiting the exact solvability of the conformal field theory. The Hilbert space is truncated such that only states having energy lower than a given cutoff $E_{\text{cut}}$ are kept. The truncation is carried out on the level of the conformal field theory spectrum where it is parameterised by the maximal conformal level,

$$N_{\text{cut}} = \frac{R}{2\pi} E_{\text{cut}} . \tag{2.5}$$

We first determine the ground state of the Hamiltonian with $h = h_i$ and then perform the time evolution by the post-quench Hamiltonian with $h = h_f$. The TCSA computations are carried out in volume $R$ with periodic boundary conditions. We measure dimensionful quantities (e.g. system size, time, operator expectation values) in units of a mass gap which for Type I quenches is the post-quench mass $m_1$. For Type II quenches the post-quench mass is not known exactly so we use the pre-quench mass $m_1^{(0)}$. For Type I quenches, for example, the volume is thus parameterised by the dimensionless quantity $r = m_1 R$, time is measured in units of $m_1^{-1}$, while operator expectation values are rendered dimensionless by appropriate powers of $m_1$. The cutoff dependence is eliminated by performing the calculations at different energy cutoffs and then extrapolating to infinite cutoff.

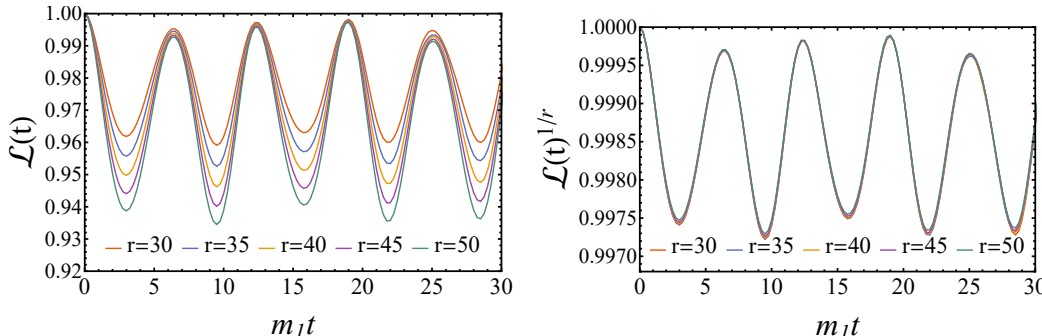

Figure 3.1: Loschmidt echo $\mathcal{L}(t)$ after a quench of size $\xi = (h_i - h_f)/h_f = 0.5$ at five different volumes in the range $r = 30 \dots 50$. Volume dependence is almost completely absent for $\mathcal{L}(t)^{1/r}$. Time and volume is measured in units of the inverse mass $m_1^{-1}$ of the lightest particle, $r = m_1 R$.

# 3 Quenches preserving integrability

First we focus on quenches in the first class in which both the pre-quench and post-quench Hamiltonian correspond to the integrable $E_8$ model at different values of the magnetic field.

## 3.1 Loschmidt echo and absence of relaxation

The Loschmidt echo

$$\mathcal{L}(t) = |\langle \Psi(t)|\Psi(0)\rangle|^2 \propto e^{-r\ell(t)}, \tag{3.1}$$

with $\langle \Psi(0)|$ denoting the pre-quench ground state and $|\Psi(t)\rangle = \exp(-\imath H t)|\Psi(0)\rangle$, measures the overlap of the time-evolving state with the initial state. Due to translational invariance, the Loschmidt echo depends exponentially on the volume parameterised by $r$, and $\ell(t)$ is a function which displays no significant volume dependence up to times $t < R/2$. For $t \geq R/2$, the fastest quasi-particles released by the quench in opposite spatial directions start encountering each other due to the periodic boundary conditions (we use units in which the maximum propagation velocity is 1). In such a finite system the Loschmidt echo shows a revival phenomenon, i.e. after sufficiently long time the state comes close to the initial state and the Loschmidt echo takes values close to 1. The corresponding timescale can be extremely large, but partial revivals are expected after some time that is of the order of the volume of the system, corresponding to quasi-particles created in the quench going around the volume and reappearing at their original position. Viewed from the thermodynamical limit these are obviously finite volume artifacts, and our simulations ran up to times short enough to exclude these revivals.

Fig. 3.1 shows the Loschmidt echo extracted from the TCSA simulation for a quench of size $\xi = 0.5$ corresponding to the relation $h_i = 1.5 h_f$. To eliminate volume dependence we also plot the Loschmidt echo (3.1) to the power $1/r$. The data display neither partial nor full revivals; revivals would occur at times depending on the volume, which would manifest in the curves in Fig. 3.1 (b) at different volumes ceasing to overlap. The almost perfect overlap of the curves also shows that other possible finite size effects are negligible, too. Note that the oscillations in the Loschmidt echo exhibit no damping which means that the system does not equilibrate on the time scales considered here.

## 3.2 Two analytic approaches to time evolution

The TCSA simulations can be compared to two different approaches to compute time evolution using field theory methods. The *perturbative quench expansion* approach introduced by Delfino [51] assumes that the quench starts from an integrable Hamiltonian $H_0$, which is changed during the quench by the addition of an extra local interaction to

$$H = H_0 + \lambda \int dx \, \Psi(x), \tag{3.2}$$

where $\lambda$ is small enough to justify the application of perturbation theory. The post-quench Hamiltonian $H$ does not need not be integrable. To first order in $\lambda$ and including only one-particle contributions, the perturbative prediction for the post-quench time evolution of a local operator $\Phi$ is [52]

$$\langle \Phi(t) \rangle = \langle 0|\Phi|0 \rangle + \lambda \sum_{i=1}^{8} \frac{2}{\left( m_i^{(0)} \right)^2} F_i^{(0)\Psi*} F_i^{(0)\Phi} \cos\left( m_i^{(0)} t \right) + \cdots + C_\Phi, \tag{3.3}$$

where $|0\rangle$ is the pre-quench vacuum, $m_i^{(0)}$ are pre-quench one-particle masses and $C_\Phi$ is included to satisfy the initial condition that the $\langle \Phi(t) \rangle$ function is continuous at $t = 0$. The amplitudes $F_{1,i}^{(0)\Phi}$ are the one-particle form factors of the $\Phi$ operator

$$F_i^{(0)\Phi} = \langle 0|\Phi|A_i(0)\rangle^{(0)} \tag{3.4}$$

between the vacuum and a one-particle state $|A_i(0)\rangle^{(0)}$ containing a single excitation of species $i$ with zero momentum. Superscript (0) means that quantities are taken at their pre-quench values which affects also the normalisation of form factors (for details of form factor normalisation see Appendix A).

The ellipsis denote the contribution of higher particle states; their omission corresponds to a low-energy approximation valid for long enough times $t \gtrsim 1/m_1^{(0)}$. Validity of perturbation theory for the time evolution operator also places a theoretical upper time limit for the validity of this expression in terms of the quench amplitude

$$t^* = \lambda^{-1/(2-2h_\Psi)}, \tag{3.5}$$

where $h_\Psi$ is the conformal weight of the quenching $\Psi$ operator.

The other method, introduced by Schuricht and Essler [49] and developed further in Ref. [50, 53], builds upon the premise that the post-quench system is integrable. We shall refer to this method as the *post-quench expansion approach*. It does not assume that the quench is perturbative and therefore it has no upper time limit, while due to being a low-energy expansion it has a lower time limit in terms of the post-quench mass $m_1$. In addition, it can be considered as a systematic expansion in the post-quench particle density as a small parameter, which means that it is limited to small enough quenches, which are, however, not necessarily perturbative in the Hamiltonian sense used above. Adapting the results of [53] to the case of the $E_8$ model, the following time evolution is obtained for operator $\Phi$ to leading order:

$$\langle \Phi(t) \rangle = \langle \Omega|\Phi|\Omega \rangle + \sum_{i=1}^{8} \frac{|g_i|^2}{4} \mathrm{Re}[F_{ii}^\Phi(\iota\pi,0)] + \sum_{i=1}^{8} \mathrm{Re}[g_i F_i^\Phi e^{-\iota m_i t}]$$

$$+ \sum_{i \neq j} \mathrm{Re}\left[ \frac{g_i^\star g_j}{2} F_{ij}^\Phi(\iota\pi,0) e^{-\iota(m_j - m_i)t} \right] + \dots, \tag{3.6}$$

where $|\Omega\rangle$ is the post-quench vacuum state,

$$\frac{g_i}{2} = \langle\Psi(0)|A_i(0)\rangle \tag{3.7}$$

is the overlap of the initial state $|\Psi(0)\rangle$ with a zero-momentum post-quench one-particle state of species $i$, and the $F^\Phi$ form factors are the matrix elements

$$F_i^\Phi = \langle\Omega|\Phi|A_i(0)\rangle \,, \qquad F_{ij}^\Phi(\vartheta_1,\vartheta_2) = \langle\Omega|\Phi|A_i(\vartheta_1)A_j(\vartheta_2)\rangle \,. \tag{3.8}$$

Note that in Eq. (3.6) $F_{ij}^\Phi(\iota\pi,0) = \langle A_i(0)|\Phi|A_j(0)\rangle$. Form factors of the operators $\sigma$ and $\epsilon$ for the lowest lying states were constructed in Ref. [84]. Unlike the perturbative approach, the method does not construct the initial state but needs the one-particle overlaps $g_i$ as inputs. We determined them by explicitly constructing the corresponding states in TCSA and computing their scalar products with the initial state.

Once again, the ellipsis indicate the contribution involving higher many-particle states. Such higher order contributions contain secular terms proportional to powers of $t$, and their resummation may lead to functions $\sim e^{\iota at}$ and $\sim e^{-bt}$. The first one corresponds to frequency shifts, while the second one is a decay factor implying a finite life-time of the oscillations appearing in Eq. (3.6). Both effects are consequences of the finite post-quench particle densities. Since the TCSA data for the relevant quenches show no such damping on the time-scales of our simulations (see below), we neglected such terms in our considerations. Their computation requires the knowledge of two-particle overlaps discussed below in Section 5; in principle they can be obtained from TCSA, but presently the limited information we can extract about them is not sufficient to evaluate the related predictions of [53].

The post-quench expansion eventually implements the standard picture of a quench starting from an initial state $|\Psi(0)\rangle$ and evolving with a post-quench Hamiltonian $H$, where the expectation values of an observable $\mathcal{O}$ can be expressed as

$$\langle\mathcal{O}(t)\rangle = \sum_{k,l} \langle\Psi(0)|\phi_k\rangle \langle\phi_k|\mathcal{O}|\phi_l\rangle \langle\phi_l|\Psi(0)\rangle \, e^{-\iota(E_l - E_k)t}, \tag{3.9}$$

with $|\phi_k\rangle$ denoting the eigenstates of the post-quench Hamiltonian $H$ with eigenvalues $E_k$. The long time average of the observable is given by the diagonal ensemble average

$$\overline{\langle\mathcal{O}(t)\rangle} = \langle\mathcal{O}\rangle_D = \sum_k |\langle\Psi(0)|\phi_k\rangle|^2 \langle\phi_k|\mathcal{O}|\phi_k\rangle \,. \tag{3.10}$$

## 3.3 Comparison with TCSA

In this section we compare our numerical results with the predictions of the two analytic approaches introduced in the previous section.

### 3.3.1 $\sigma$ operator

The $\sigma$ operator is perhaps the most important physical observable of the Ising field theory, since it corresponds to the continuum limit of the spin chain magnetisation. We now present comparison of Eqs. (3.3) and (3.6) in terms of the time evolution of the magnetisation.

Specifically, the perturbative prediction Eq. (3.3) for the time evolution of the magnetisation gives

$$\langle\sigma(t)\rangle = \langle0|\sigma|0\rangle + \lambda \sum_{i=1}^8 \frac{2}{\left(m_i^{(0)}\right)^2} \left|F_i^{(0)\sigma}\right|^2 \cos\left(m_i^{(0)}t\right) + \cdots + C_\sigma \,. \tag{3.11}$$

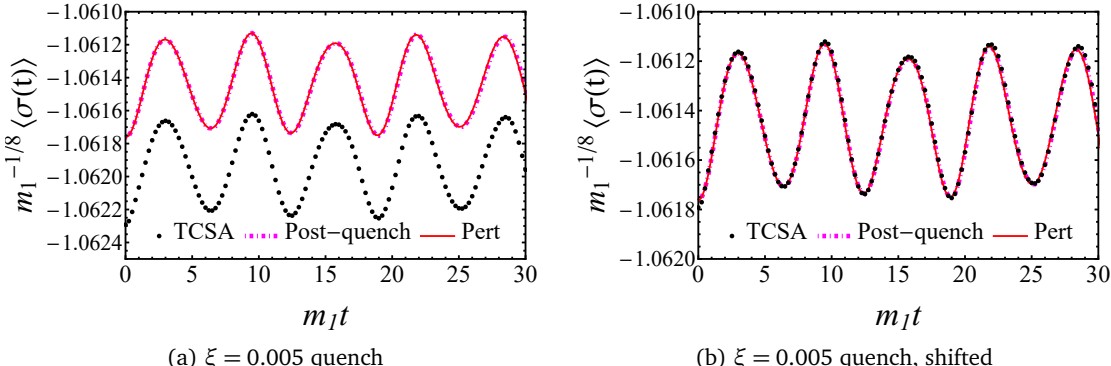

(a) $\xi = 0.005$ quench

(b) $\xi = 0.005$ quench, shifted

Figure 3.2: Time evolution of the $\sigma$ operator after a small quench of size $\xi = (h_i - h_f)/h_f = 0.005$. Comparison of the TCSA data (black dots) with the first order perturbative quench expansion result (3.11),(3.12) (red line) and the prediction of the post-quench expansion (3.13) (magenta dashed line). In panel (b) the curves are shifted on top of each other. TCSA data are for volume $r = 40$ and are extrapolated in the truncation level. Time is measured in units of the inverse mass $m_1^{-1}$ of the lightest particle. Expectation values are measured in units of $m_1^{1/8}$.

The constant $C_\sigma$ is given in Ref. [52] as an infinite form factor series. A self-consistent choice in the spirit of the derivation of this constant is to truncate its series at the one-particle term, this guarantees the continuity of $\langle \sigma(t) \rangle$ at $t = 0$ at this order of the form factor expansion. However, the quench under consideration falls in the class for which the form factor series for $C_\sigma$ can be summed up with the result

$$C_\sigma = \frac{1}{15} \frac{h_f - h_i}{h_i} \langle 0|\sigma|0\rangle \,. \tag{3.12}$$

Eq. (3.12) should give a more accurate prediction for the baseline of the oscillations so we use this expression for $C_\sigma$ in this section.

For the post-quench method we obtain from Eq. (3.6)

$$\langle \sigma(t) \rangle = \langle \Omega|\sigma|\Omega \rangle + \sum_{i=1}^{8} \frac{|g_i|^2}{4} \mathrm{Re}[F_{ii}^\sigma(\imath\pi, 0)] + \sum_{i=1}^{8} \mathrm{Re}[g_i F_i^\sigma e^{-\imath m_i t}]$$
$$+ \sum_{i \neq j} \mathrm{Re}\left[ \frac{g_i^\star g_j}{2} F_{ij}^\sigma(\imath\pi, 0) e^{-\imath(m_j - m_i)t} \right] + \dots \,. \tag{3.13}$$

In both approaches we use the exact infinite volume results for the operator expectation values [85].

The comparison between the TCSA simulations and the analytic expansions is shown in Figs. 3.2-3.5. In principle, the TCSA data have a residual error resulting from cutoff extrapolation, but the cutoff-dependence is mainly restricted to a shift in the time-independent baseline of oscillations. As for the oscillations, their error is comparable to the linewidth of Fig 3.2. Since the baseline is adjusted for comparison (see below), we decided to omit error bars in our plots. The volume dependence is similarly absent: the difference between curves measured at $r = 30$ and $r = 50$ is so tiny that it is completely invisible. As a result, our simulation results can be interpreted as being identical to the physical ones for infinite volume and cutoff for the time window shown. For details of TCSA cutoff extrapolation and volume dependence cf. Appendix B.2.

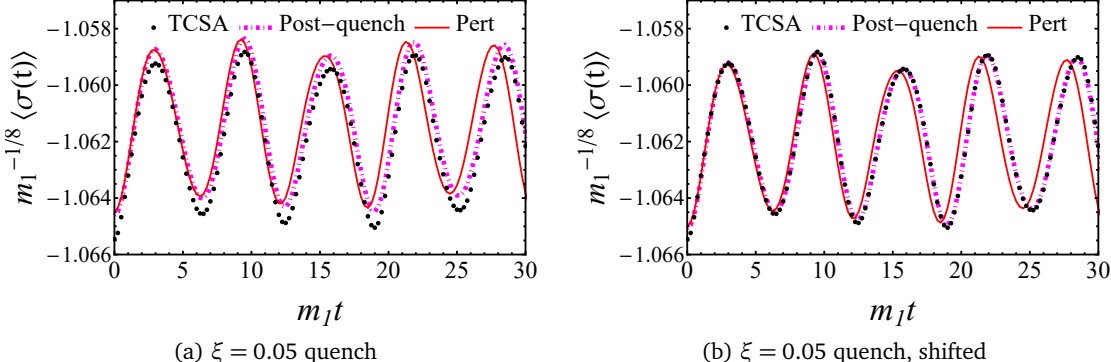

(a) $\xi = 0.05$ quench

(b) $\xi = 0.05$ quench, shifted

Figure 3.3: Time evolution of the $\sigma$ operator after a small quench of size $\xi = 0.05$. In panel (b) both theoretical results are shifted to the diagonal ensemble value obtained from TCSA. TCSA data are for volume $r = 40$ and are extrapolated in the truncation level. Notations and units are as in Fig. 3.2.

We start our systematic analysis with very small quenches and move towards larger quenches afterwards. Fig. 3.2a shows the time evolution of the $\sigma$ operator after a quench of size $\xi = 0.005$. The two analytic approaches give identical results which however differ from the TCSA data. The deviation is well within the error bar of the TCSA method (note the scale on the $y$-axis!). The main source of error is the cutoff extrapolation which manifests itself as a time-independent shift in the expectation value. In Fig. 3.2b we shifted the curves on top of each other to show that for such a small quench there is perfect agreement between the two analytic approaches and, up to a tiny constant, with the numerical data.

Let us now consider a somewhat larger quench of size $\xi = 0.05$. The predictions of the analytic approaches are shown in Fig. 3.3a. Both approaches seem to deviate from the numerical result roughly to the same amount which is well beyond the TCSA error. The main part of the deviation is a constant shift with respect to the TCSA data.

In both approaches, the oscillations are centred at the respective time averages given by the time independent terms in Eq. (3.11) and Eq. (3.13). It is clear that these terms in Eq. (3.13) correspond to the lowest order approximation of the diagonal ensemble expectation value, and that summing up all time-independent terms would eventually compute the full diagonal average which is indeed expected to give the baseline of the oscillations. For the perturbative approach corrections to $C_\sigma$ would come from higher orders in $\lambda$.

The diagonal average can be easily computed in the TCSA framework, which allows us to implement a modified version of both approaches by replacing their respective time independent terms by the numerically determined diagonal average. This is plotted in Fig. 3.3b which shows that while the post-quench expansion supplemented by the numerical overlaps can be brought to perfect agreement with the TCSA data by a constant shift, the first order perturbative quench expansion shows a frequency mismatch. As the omitted higher form factors in the series are expected to modify the short time behaviour and vanish for large time, the discrepancy indicates that this quench is beyond the domain of validity of first order perturbation theory.

We recall that the post-quench approach is not fully analytic as it needs the $g_i$ overlaps as inputs. As the amplitudes of the oscillation in Fig. 3.3b set by these overlaps are close in the different approaches, the better agreement between numerics and the post-quench approach does not come from a more precise knowledge of the overlaps but from the fact that the post-quench expansion predicts oscillations with frequencies set by the post-quench particle masses, while the perturbative expansion uses the pre-quench frequencies in the time evolution.

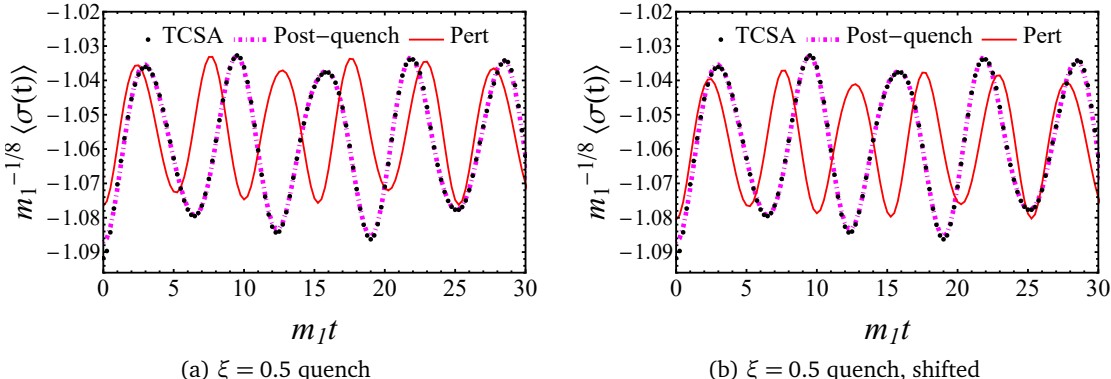

(a) $\xi = 0.5$ quench

(b) $\xi = 0.5$ quench, shifted

Figure 3.4: Time evolution of the $\sigma$ operator after a quenche of size $\xi = 0.5$. In panel (b) both theoretical results are shifted to the diagonal ensemble value obtained from TCSA. TCSA data are for volume $r = 40$ and are extrapolated in the truncation level. Notations and units are as in Fig. 3.2.

For this quench the time range of validity of the perturbative approach in Eq. (3.5) is $t^* = |h_i - h_f|^{-8/15}$ which in terms of our dimensionless time used in the plots is $m_1 t^* \approx 4.4 \xi^{-8/15}$. For $\xi = 0.05$ we get $t^* \approx 20$, whose order of magnitude agrees with the range of good agreement seen in Fig. 3.3b. Note however, that without shifting the baseline of oscillations the agreement is worse, see Fig. 3.3a. The reason is that in Eq. (3.11) the form factor series is truncated at the one-particle term which is not the full first order perturbative result. As summation of the series is not possible, in practical calculations a finite number of terms are kept.

It is also instructive to consider a much larger quench with $\xi = 0.5$ which is shown in Fig. 3.4. Notice that unlike the first order perturbative result (3.11), the post-quench expansion (3.13) agrees very well with the TCSA time evolution even without the shift, so the constant terms in Eq. (3.13) already give a decent approximation of the diagonal average. The excellent agreement also implies that this quench is still in the low density regime. In this case apart from the frequency the amplitudes are also in disagreement with the first order perturbative prediction, but this is no surprise since this quench is well beyond the perturbative regime.

### 3.3.2 $\epsilon$ operator

The scaling Ising field theory has another relevant operator, the field $\epsilon$ with conformal weight $h_\epsilon = 1/2$ which corresponds to the transverse magnetisation in the spin chain.

The comparison of the two approaches with the TCSA results is shown in Fig. 3.5. Similarly to the case of the $\sigma$ operator, for a $\xi = 0.05$ quench the perturbative prediction, once it is shifted to the TCSA diagonal average, agrees well with the numerical result up to $m_1 t \approx 10$ after which the frequency mismatch causes deviations. The quench of size $\xi = 0.5$ is outside of the perturbative domain, but the post-quench expansion agrees well with the TCSA data even without the constant shift to the diagonal average.

### 3.4 Fourier spectra of the post-quench time evolution

Based on Eq. (3.6), the time evolutions $\langle \sigma(t) \rangle$ and $\langle \epsilon(t) \rangle$ are expected to be dominated by frequencies corresponding to the post-quench particle masses. In addition, one also expects the presence of mass differences in the spectrum (cf. Eq.(3.9)). However, the overlap factors $g_i$ are of order $\xi$, so while the amplitudes of the frequencies $m_i$ are of order $\xi$, those of $m_i - m_j$

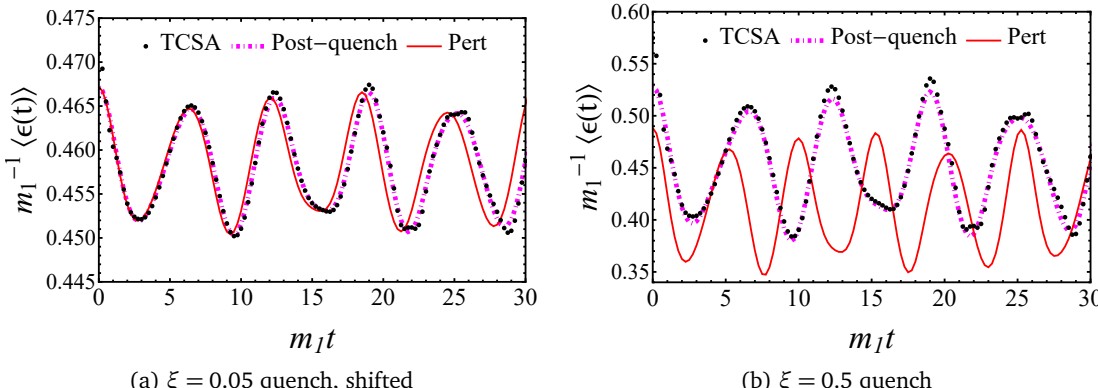

(a) $\xi = 0.05$ quench, shifted             (b) $\xi = 0.5$ quench

Figure 3.5: Time evolution of the $\epsilon$ operator after a quench of size (a) $\xi = 0.05$ and (b) $\xi = 0.5$ In panel (a) both theoretical results are shifted to the diagonal ensemble value obtained from TCSA. TCSA data are for volume $r = 40$ and are extrapolated in the truncation level. Notations and units are as in Fig. 3.2. Expectation values are measured in units of $m_1$.

are of order $\xi^2$. Indeed the Fourier spectra in Fig. 3.6 show prominent peaks at the positions of the eight particle masses, with a much smaller peak at $m_2 - m_1$. Fig. 3.6 also shows the Fourier spectrum of the Loschmidt echo; it is notable that the difference peak is absent.

The fact that the qualitative behaviour of the different operators is roughly the same helps identify one-particle peaks. Although the $E_8$ spectrum consists of eight particles, only the first three of those can be seen clearly in Fig. 3.6. The rest are above the two-particle threshold and are hard to distinguish against the background of many-particle states.

Closer examination shows that the one-particle peaks are shifted a little to the right, especially for the second and third one. This is known to originate from frequency shifts due to the finite post-quench particle density (cf. the discussion in Subsection 3.2). The resulting frequency shift almost raises the position of the third particle above the two-particle threshold, which is the reason why it is less observable than the first two.

Also note that the prominence of peaks depends on the observable, which can be understood from Eq. (3.9): in addition to the overlap factors, their height also depends on the operator matrix element (form factor) associated to the given observable.

Note that for the Loschmidt echo (3.1) the peak at the mass difference frequency is not visible, which can be understood by the following reasoning. The Loschmidt echo is expressed by a square of a scalar product (cf. Eq. (3.1)) implying that the coefficients in its Fourier spectrum contain a maximum of four $g_i$ overlap factors. In case of one-particle frequencies, the coefficient has two $g_i$ factors, which is one more compared to the case of an operator time-evolution (cf. Eq. (3.13)). However, in a term having a frequency equal to a mass difference the coefficient contains four $g$ factors that is two more compared to that of an operator. Since the overlaps are small, higher order contributions are suppressed, leading to the absence of the mass difference frequency peak in the Loschmidt spectrum.

The erratic behaviour, consisting of a jagged landscape of multiple Fourier peaks right above the two-particle threshold $\omega = 2$ is due to two effects. First, there are five stable one-particle states in the $E_8$ spectrum above the two-particle threshold that correspond to localised peaks; the $E_8$ masses are indicated by the continuous (black) vertical lines in Fig. 3.6. Second, the spectrum of multi-particle levels in a finite volume is also discrete, corresponding to isolated peaks in the spectrum, which acquire a non-zero width in the finite density post-quench environment. The spacing between these levels decreases with increasing $\omega$, therefore

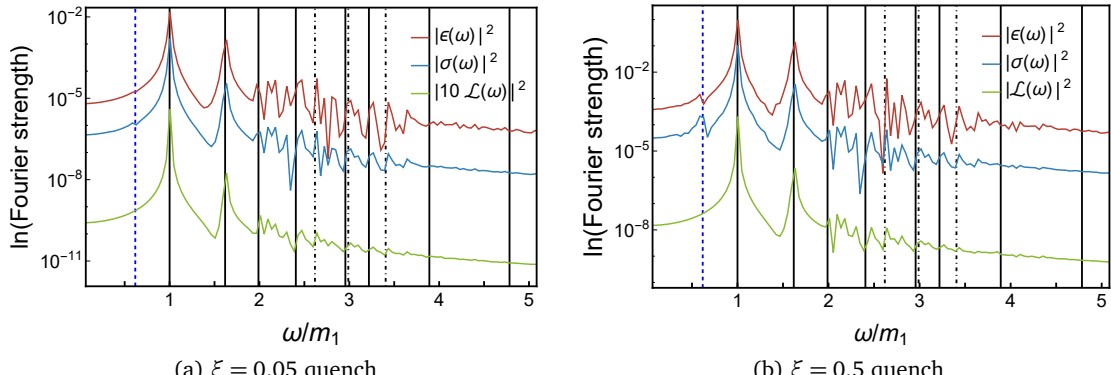

(a) $\xi = 0.05$ quench    (b) $\xi = 0.5$ quench

Figure 3.6: Fourier spectra of the time evolved operators and the Loschmidt echo for quenches of size (a) $\xi = 0.05$ and (b) $\xi = 0.5$ in volume $r = 50$. Continuous gridlines are at single particle masses of the $E_8$ spectrum, green lines are at sums of two particle masses. The blue (leftmost) continuous line is located at the difference of the first two masses. The frequency $\omega$ is measured in units of the mass of the lightest particle $m_1$, operator expectation values are measured in units of appropriate powers of $m_1$.

the Fourier spectrum eventually becomes smooth for larger $\omega$. In addition, the level widths are expected to increase with the quench size, so the Fourier spectrum is expected to become smoother for larger quenches, which is confirmed by the data displayed in Fig. 3.6.

The observed Fourier spectra provide a partial explanation of the deviation of the first order perturbative predictions from the eventual time evolution. As we see, the post-quench evolution is dominated by the post-quench frequencies, while the expansion (3.3) contains the pre-quench frequencies.

## 4 Integrability breaking quenches

Let us now turn to quenches that break integrability. Non-integrable systems, in contrast to integrable ones, are generally thought to thermalise in the sense of locally equilibrating to a thermal Gibbs ensemble. It is an interesting question which features of the time evolution observed for the integrable case are robust under integrability breaking. A related and interesting question is whether there is any sign of relaxation, i.e. damping of the oscillations, which was notably absent for integrable quenches, at least for the time scales accessible for the TCSA simulation.

Non-integrable quenches in the vicinity of the axis $h = 0$ were already studied in Ref. [65]; these correspond to a small breaking of integrability of the free massive Majorana theory. Here we look at the opposite limit keeping $h$ finite and fixed, and quenching by switching on a nonzero value of $M$, which corresponds to breaking integrability of the $E_8$ theory. This can be done in two ways: either towards the ferromagnetic or the paramagnetic phase, depending on whether the sign of $M$ is positive/negative, respectively. These quenches can be described in terms of the dimensionless parameter $\eta$ (cf. eqn.(2.1)):

$$\eta = \frac{M}{|h|^{8/15}}. \tag{4.1}$$

Note that since the post-quench theory is non-integrable, the post-quench series (3.6) cannot be applied and the only analytical prediction comes from Delfino's perturbative expansion (3.3).

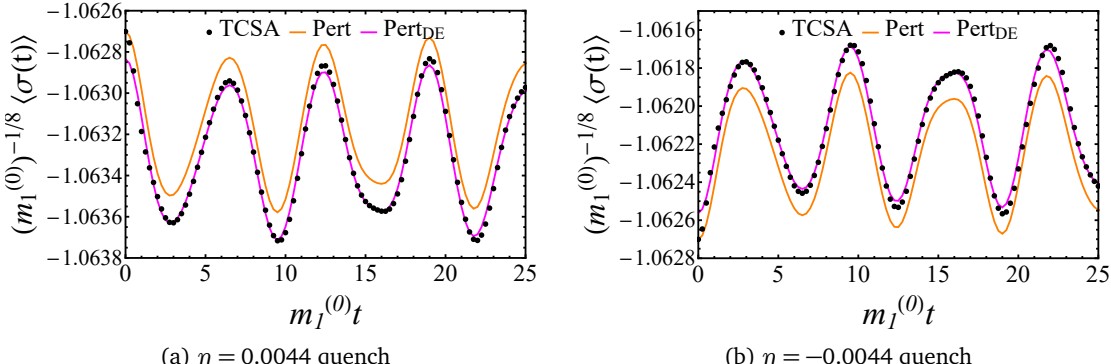

(a) $\eta = 0.0044$ quench

(b) $\eta = -0.0044$ quench

Figure 4.1: Plot of $\langle \sigma(t) \rangle$ after quenches of size $|\eta| = 0.0044$. The blue dots are the TCSA results, the prediction of the perturbative quench expansion is shown in orange lines (see text for details), while the magenta lines represent the perturbative prediction shifted to the diagonal ensemble average. TCSA data are for volume $r = 50$ and are extrapolated in the truncation level. Time is measured in units of the pre-quench mass of the lightest particle $m_1^{(0)}$. Note that it coincides with $m_1$ of Sec. 3.

The spectrum of the post-quench theory is well understood in both regimes. In the immediate vicinity of the axis only 3 stable particles remain as the other five are over the two-particle threshold and since integrability does not protect them, they have a finite life-time and decay [84, 86].

For larger values of $M$, however, the behaviours in the two regimes are markedly different. In the ferromagnetic regime there appear mesons whose number increases with the magnitude of $M$. When approaching the thermal axis ($\eta = \infty$), the related poles in the scattering matrix fuse together to form a branch cut corresponding to a continuum of two-kink states under the so-called McCoy–Wu scenario [87]. Close to the $h = 0$ axis the physics corresponds to the confinement of kink states which results in interesting effects on the time evolution [88]. The meson spectrum resulting from the confinement in the vicinity of thermal axis is well-understood [89–91], while close to the $E_8$ axis $M = 0$ the particle spectrum can be described using form factor perturbation theory developed in Ref. [92].

In the paramagnetic regime, with increasing magnitude of $\eta$ the number of stable particles is first reduced to two and then to one. The threshold values for the decays of the third and second particles are $\eta_3 = -0.138$ and $\eta_2 = -2.08$, respectively [93].

## 4.1 Small quenches

We start with very small quenches in both directions, choosing $|\eta| = 0.0044$ to be safely in the perturbative regime, so that we can compare the TCSA results with the prediction of the perturbative quench expansion. In this case Eq. (3.3) predicts the following evolution for the magnetisation:

$$\langle \sigma(t) \rangle = \langle 0|\sigma|0 \rangle + \lambda \sum_{i=1}^{8} \frac{2}{\left(m_i^{(0)}\right)^2} F_i^{(0)\epsilon *} F_i^{(0)\sigma} \cos\left(m_i^{(0)} t\right) + \cdots + \tilde{C}_\sigma \,, \qquad (4.2)$$

where

$$\lambda = -\frac{M}{2\pi} = -\frac{\eta |h|^{8/15}}{2\pi} \,. \qquad (4.3)$$

For this quench there is no exact result for $\tilde{C}_\sigma$ similar to Eq. (3.12) so we keep the one-particle term in its form factor expansion as for the time-dependent part.

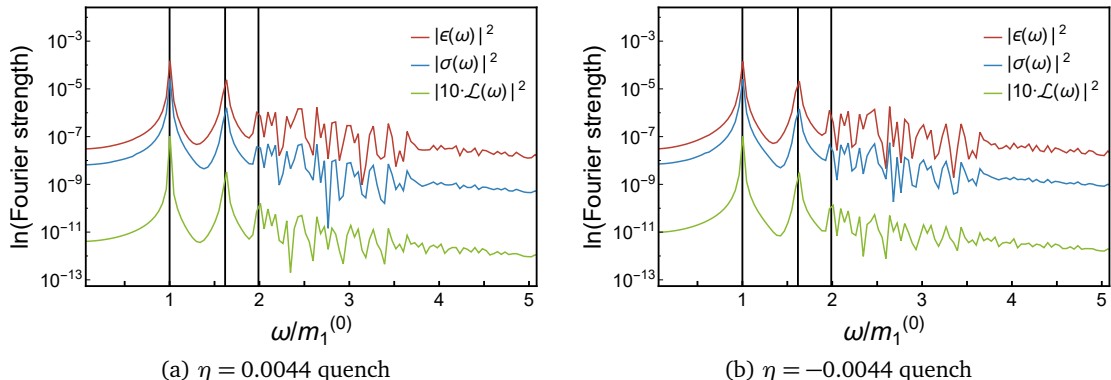

(a) $\eta = 0.0044$ quench  (b) $\eta = -0.0044$ quench

Figure 4.2: Fourier spectra after quenches of size $\eta = |0.0044|$ in volume $r = 50$. The Fourier amplitude of the Loschmidt echo is magnified for convenience. Continuous gridlines refer to the three stable particle masses given in Eq. (4.4). Frequency is measured in units of $m_1^{(0)}$.

Note that the prediction is symmetric in the two directions apart from a relative sign in the oscillations, despite the fact that the physics essentially differs in the ferromagnetic/paramagnetic domains. This is a feature of first order perturbation theory. The numerical simulation is consistent with this behaviour demonstrating that quenches of this size are truly in the perturbative domain. As Fig. 4.1 shows, the TCSA result differs from the first order result (4.2) by a constant shift, but if one applies the correction to the diagonal ensemble average, there is excellent agreement between the two apart from a very short initial transient. Note that there is no sign of damping of one-particle oscillations at the time scale of the simulation.

The Fourier spectra of the time evolution are shown in Fig. 4.2, and they are very similar for both signs of the coupling. The first three peaks can be clearly identified with the masses of the three stable particle excitations, which can be calculated perturbatively using form factor perturbation theory [92]:

$$m_i \simeq m_i^{(0)} + M \frac{F_{ii}^{\epsilon(0)}(\iota\pi, 0)}{m_i}, \qquad i = 1, \dots, 8, \tag{4.4}$$

where $m_i^{(0)}$ is the mass of the $i$th particle in the $E_8$ model. The correction $m_i - m_i^{(0)}$ is too small to be visible in our plots. Our results show that for very small quenches away from the $E_8$ axis the perturbative expansion gives a very good approximation in a reasonably wide time window.

## 4.2 Larger quenches in paramagnetic direction

Let us now consider larger quenches, first in the paramagnetic direction to see how the picture changes reflecting the physics of this particular phase.

### 4.2.1 $\eta = -0.125$ quench

The time evolution of the magnetisation is shown in Fig. 4.3a. The agreement with the first order perturbative expansion involving the pre-quench frequencies is now less satisfactory because the difference between pre- and post-quench frequencies becomes visible. Apart from this, there is a difference in the amplitudes as well, which is shown in the disagreement between TCSA data and the perturbative prediction shifted to the diagonal ensemble average.

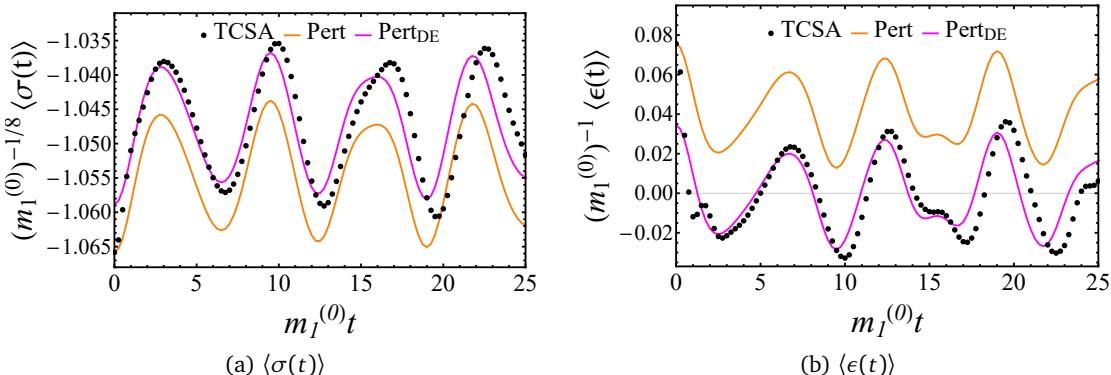

Figure 4.3: Time evolution of the (a) $\sigma$ and (b) $\epsilon$ operator after a quench of size $\eta = -0.125$. TCSA data are for volume $r = 50$ and are extrapolated in the truncation level. Notations and units are as in Fig. 4.1.

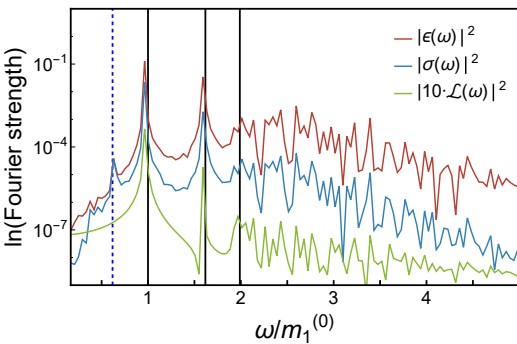

Figure 4.4: Fourier spectra $\eta = -0.125$ quench, $r = 50$. The vertical lines show the masses predicted by (4.4), the blue line is drawn at the $m_2 - m_1$ difference frequency. Units are as in Fig. 4.2.

Temporal fluctuations of $\langle \epsilon(t) \rangle$ plotted in Fig. 4.3b show similar difference between the first order perturbative result and TCSA[1].

For these quenches the time scale (3.5) is $m_1^{(0)} t^* = m_1^{(0)}/|\lambda| \approx 4.4 \cdot 2\pi/|\eta|$ which for $\eta = 0.125$ gives $m_1^{(0)} t^* \approx 220$. In contrast to this, deviations from the numerical data are clearly visible at times that are at least an order of magnitude smaller than $t^*$. Including more terms in the form factor series would lead to a better agreement but mainly for short times at the order of $m_1^{(0)} t \sim 1$. We return to the origin of this mismatch in Sec. 6.

At this value of the coupling, the third particle is still stable but it is very close to the threshold of instability which is indeed reflected in the Fourier spectra displayed in Fig. 4.4. The vertical lines show the masses predicted by form factor perturbation theory in Eq. (4.4). The first two peaks are visibly shifted from the first order perturbative value which can be due to higher order corrections, but also to additional frequency shifts from the finite post-quench density that are predicted by the post-quench expansion formalism. Also note the appearance of Fourier peaks corresponding to mass differences; the expected position of the dominant one is indicated by the leftmost vertical blue line.

---

[1] We remark that for non-integrable quenches, due to the presence of the Hamiltonian perturbation $\epsilon$ the expectation value $\langle \epsilon(t) \rangle$ diverges logarithmically with the cutoff [65] and it needs to be regularised. The divergent term is proportional to the identity operator, so it merely causes a time-independent costant shift that changes logarithmically with cutoff, which is easy to compensate during the cutoff extrapolation (for details cf. Appendix B.2).

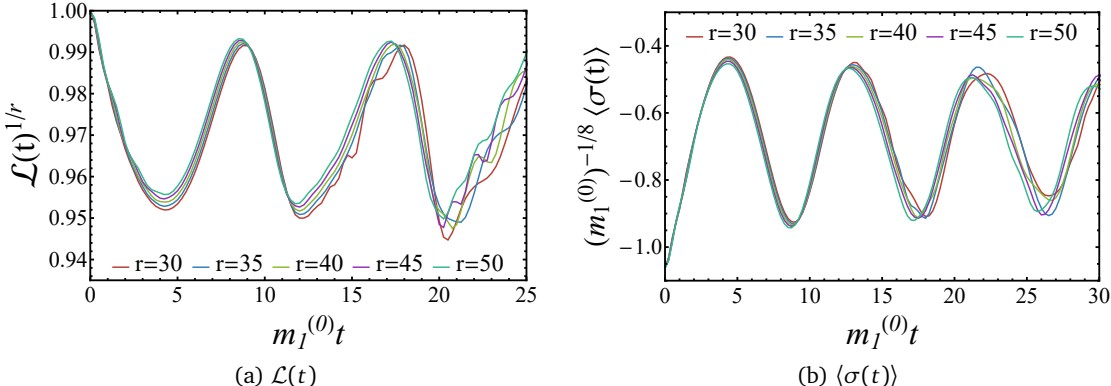

(a) $\mathcal{L}(t)$                    (b) $\langle \sigma(t) \rangle$

Figure 4.5: Time evolution of the (a) Loschmidt echo and the (b) $\sigma$ operator after a quench of size $\eta = -1.38$ in different volumes. Time is measured in units of the inverse mass of the lightest particle $\left( m_1^{(0)} \right)^{-1}$, operator expectation values are measured in units of appropriate powers of $m_1^{(0)}$.

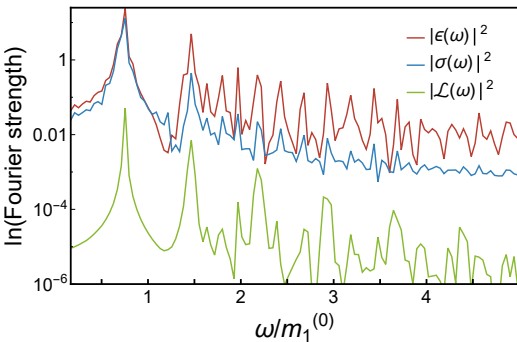

Figure 4.6: Fourier spectra of the time evolution after a quench of size $\eta = -1.38$ in volume $r = 45$. Units are as in Fig. 4.2.

### 4.2.2   $\eta = -1.38$ quench

Considering an even larger quench we get the first observation of damping which appears both in the oscillations the Loschmidt echo and the expectation value of the magnetisation $\sigma$. For this quench cutoff errors in TCSA are much larger, but extrapolation in the cutoff is still reliable. On the other hand, the numerical results now also display a visible volume dependence as shown in Fig. 4.5, in particular, the Loschmidt echo becomes quite noisy for times $t > R/2$.

Despite these limitations we can still draw some robust conclusions. The damping is clearly visible in the time evolution of the magnetisation, and is also manifested in the broadening of the quasi-particle peaks in the Fourier spectra in Fig. 4.6 compared to those in Fig. 4.4. While in a different regime of the Ising field theory the decay rate was accurately extracted from the TCSA data [65], in the present case it is unfortunately not possible to reliably estimate the relaxation time due to the limited time window in which the TCSA is valid.

The post-quench $\eta$ value for this quench is above the decay threshold of the third particle, so there are only two quasi-particle peaks in the spectrum. For a quench this large, perturbative approaches for both the time evolution and the mass shifts are unreliable, and so we have no analytic predictions to compare our simulation results with.

For even larger quenches, necessary to access the domain in which there is only a single

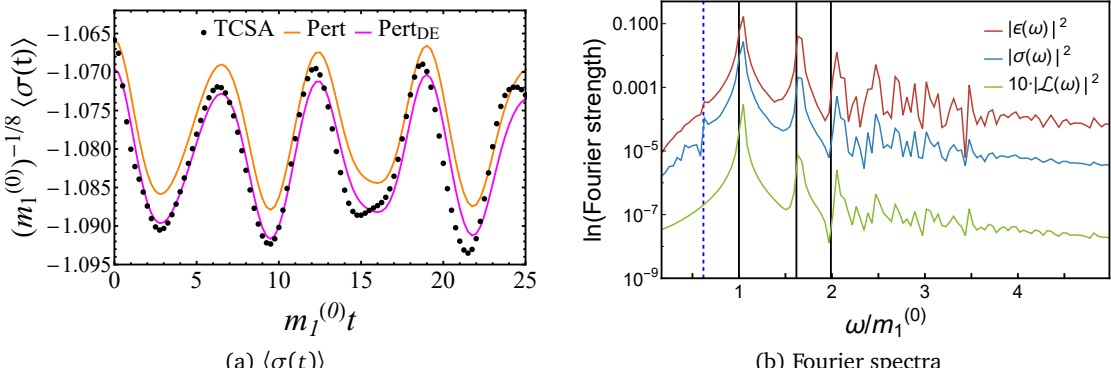

(a) $\langle \sigma(t) \rangle$         (b) Fourier spectra

Figure 4.7: Time evolution of the (a) $\sigma$ operator and (b) Fourier spectra of various quantities after a quench of size $\eta = 0.125$ in volume $r = 50$. Notations and units are as in Fig. 4.1 and Fig. 4.2.

quasi-particle, the TCSA is not convergent enough to extract any useful information and so we do not consider them here.

## 4.3 Larger quenches in ferromagnetic direction

Positive values of the quench parameter $\eta$ correspond to a final state with a ferromagnetic behaviour. In this phase, confinement of domain walls is expected to result in a characteristic spectrum of mesons. We chose the same quench amplitudes as for the paramagnetic directions in order to facilitate comparison between the two regimes.

### 4.3.1 $\eta = 0.125$ quench

At this value of $\eta$ there are only three mesons [90] which is the same as the number of stable particles in the paramagnetic direction. Contrasting the time evolution in Fig. 4.7a to Fig. 4.3a shows that the symmetry observed for very small quenches is now visibly broken. The Fourier spectra in Fig. 4.7b show three prominent quasi-particle peaks but they are significantly wider than in the paramagnetic case. Also note that the frequency shifts compared to the perturbative result (4.4) are larger and have opposite sign compared to the paramagnetic case. Apart from these differences, at this $\eta$ the two directions were comparable to each other and have shown similarities, which is not the case when $\eta$ is increased further.

### 4.3.2 $\eta = 1.38$ quench

For such a large quench the ferromagnetic case is very different from the corresponding paramagnetic quench; indeed Fig. 4.8 shows a marked qualitative difference from Fig. 4.5. There is no observable damping, however, the magnetisation shows the strong presence of a mass difference frequency.

The Fourier spectra in Fig. 4.9 display a nice regular sequence of meson excitations, which is a signal of confinement [65, 88]. The meson masses in the Ising field theory are well described by analytic methods [89–91]. However, instead of using these predictions we determined the meson masses from the TCSA spectrum of the post-quench Hamiltonian in the same volume ($r = 50$) in which the time evolution was considered, which also accounts for finite size mass corrections. As shown in Ref. [94], the TCSA meson masses agree with theoretical predictions to a high precision, so displaying the analytic results would not amount to any visible change in Fig. 4.9. Even so, the peaks in the Fourier spectra do not coincide exactly

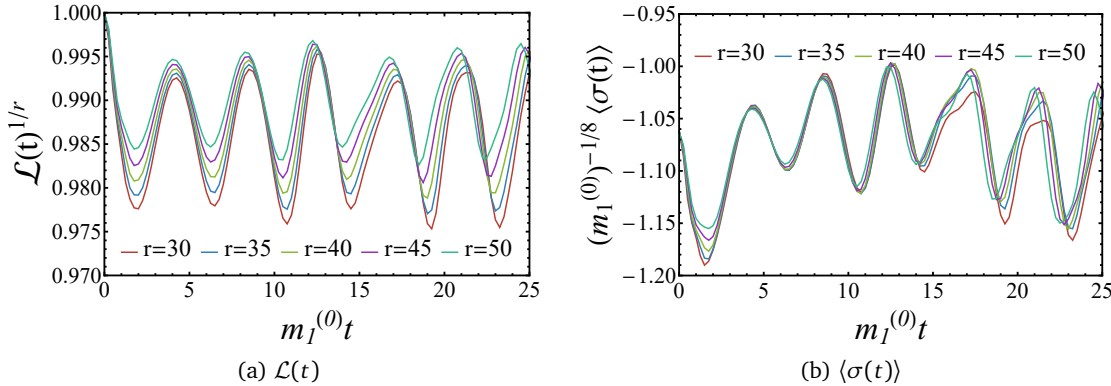

(a) $\mathcal{L}(t)$          (b) $\langle\sigma(t)\rangle$

Figure 4.8: (a) Loschmidt echo $\mathcal{L}(t)$ and (b) magnetization $\langle\sigma(t)\rangle$ after a quench of size $\eta = 1.38$ in various volumes. No damping can be observed in contrast to the paramagnetic quenches. Time is measured in units of the inverse mass of the lightest particle $\left(m_1^{(0)}\right)^{-1}$, operator expectation values are measured in units of appropriate powers of $m_1^{(0)}$.

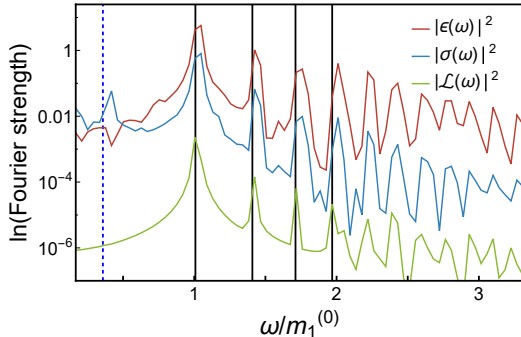

Figure 4.9: Fourier spectra for a quench of size $\eta = 1.38$ in volume $r = 50$. Meson masses are seen to get shifted due to the finite particle density. Units are as in Fig. 4.2.

with the masses, which indicates corrections due to the effect of the finite post-quench particle density.

Similarly to the paramagnetic direction, for even larger quenches the TCSA is not convergent enough to extract any useful information and so we do not consider them here.

# 5 Overlaps and statistics of work

Global quantum quenches release an extensive amount of energy into the system and result in a state of finite energy density. Assuming that the post-quench field theory is gapped, one can expand the initial state in the basis formed by asymptotic multi-particle states,

$$|\Psi(0)\rangle = \mathcal{N}\left\{|\Omega\rangle + \sum_{N=1}^{\infty}\frac{1}{N!}\left(\prod_{n=1}^{N}\int\frac{d\vartheta_n}{2\pi}\right)\tilde{K}_{i_1\ldots i_n}(\vartheta_1,\ldots,\vartheta_n)\left|A_{i_1}(\vartheta_1)\ldots A_{i_n}(\vartheta_n)\right\rangle\right\} \qquad (5.1)$$

with

$$\left|A_{i_1}(\vartheta_1)\ldots A_{i_n}(\vartheta_n)\right\rangle = A_{i_1}^{\dagger}(\vartheta_1)\ldots A_{i_n}^{\dagger}(\vartheta_n)|\Omega\rangle , \qquad (5.2)$$

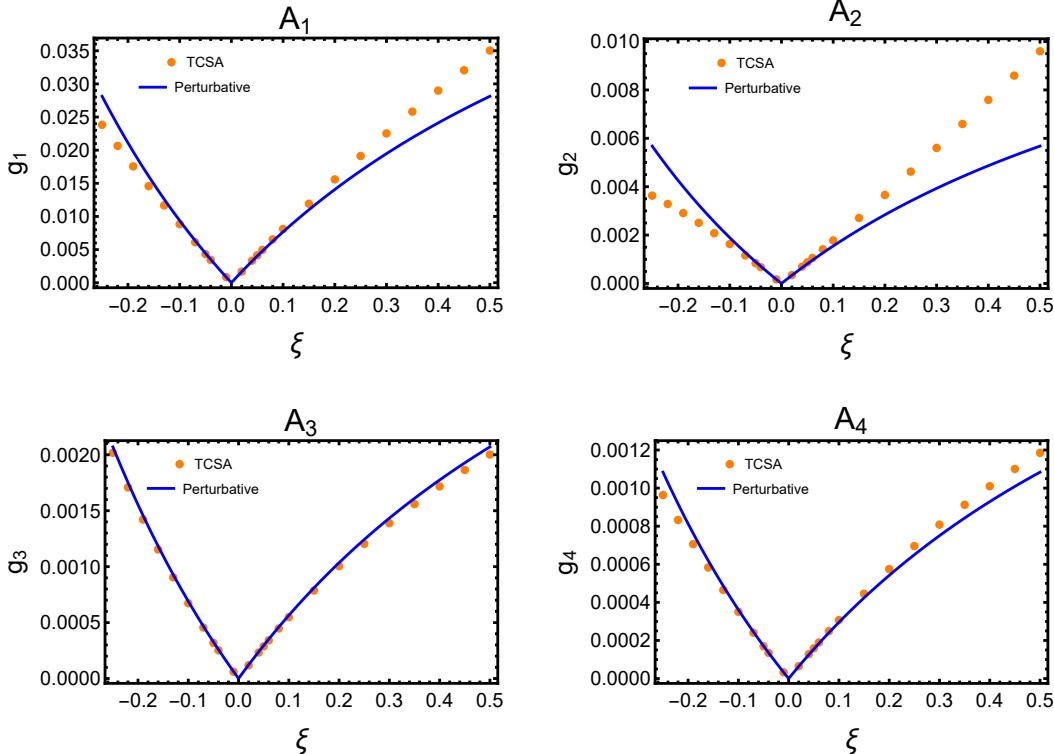

Figure 5.1: Comparison between TCSA and perturbative overlaps of the first four one-particle states as functions of the quench strength $\xi$ for quenches along the $E_8$ axis in volume $r = 40$. The dominant contribution to the time evolution is given by the first three particles, so their comparison indicates the limits of the perturbative regime.

where the operators $A_{i_1}^{\dagger}(\vartheta_1)$ satisfy the Faddeev–Zamolodchikov algebra [95, 96]:

$$A_i(\vartheta_1)A_j(\vartheta_2) = S_{ij}^{kl}(\vartheta_1 - \vartheta_2)A_k(\vartheta_2)A_l(\vartheta_1)$$
$$A_i(\vartheta_1)A_j^{\dagger}(\vartheta_2) = S_{ij}^{kl}(\vartheta_2 - \vartheta_1)A_k^{\dagger}(\vartheta_2)A_l(\vartheta_1) + \delta(\vartheta_2 - \vartheta_1)\delta_{ij}, \tag{5.3}$$

where $S_{ij}^{kl}(\theta)$ is the exact two-particle scattering matrix.

In Eq. (5.1) $|\Omega\rangle$ is the post-quench vacuum state and $\mathcal{N}$ is a normalisation factor ensuring $\langle\Psi(0)|\Psi(0)\rangle = 1$. For a global quench where both the pre-quench and post-quench Hamiltonians are translationally invariant, the overlap functions $\tilde{K}_{i_1\ldots i_n}(\vartheta_1, \ldots, \vartheta_n)$ contain as a factor a Dirac delta in the total momentum. In particular, the one-particle overlap functions $\tilde{K}_i(\vartheta)$ are related to the amplitudes $g_i$ defined in (3.7) by

$$\tilde{K}_i(\vartheta) = \frac{g_i}{2}2\pi\delta(\vartheta). \tag{5.4}$$

## 5.1 One-particle overlaps: comparing TCSA with perturbation theory

In this section we extract the one-particle overlaps from the TCSA data and compare them with the prediction of the perturbative approach.

The TCSA overlaps are extracted in finite volume so they must be converted into the infinite volume normalisation. For the case of one-particle states, the necessary normalisation factor

is [97–99]

$$\frac{g_i}{2} = \frac{\langle \Psi(0)|A_i(0)\rangle_R}{\sqrt{m_i R}}\,.$$ (5.5)

Using (3.11), the first order perturbative expression for the overlap can be read off:

$$\frac{g_{i,\mathrm{pert}}}{2} = \frac{\xi F^\sigma_{1,i}}{\left(m_i^{(0)}\right)^2}\,,$$ (5.6)

where the $m_i^{(0)}$ are the masses in the pre-quench theory, related to the post-quench masses by

$$m_i^{(0)} = m_i(1+\xi)^{1/(2-2h_\sigma)}\,,$$ (5.7)

where we used the notations of Sec. 3. The comparison is shown in Fig. 5.1. Just by looking at these results one would expect that for $|\xi| < 0.1$ the perturbative quench expansion should describe the time evolution just as well as the post-quench expansion approach. As we saw in Sec. 3, this is not the case, which can be attributed mainly to the difference between the pre-quench and post-quench frequencies.

A similar comparison can be made for the non-integrable quenches considered in Sec. 4 and is presented in Fig. 5.2. We only consider the first three particle states as they are the ones which remain stable off the $E_8$ axis. Note that in the paramagnetic direction ($\eta < 0$) the third particle overlap deviates earlier from the perturbative result, which is due to the particle becoming unstable as indicated by the comparison between $m_3$ and the two-particle threshold $2m_1$.

## 5.2 Statistics of work and two-particle overlaps

A special class of quenches is given by the so-called integrable ones, where the initial state (5.1) can be written in a generalised squeezed state form

$$|\Psi(0)\rangle = \mathcal{N}\exp\left\{\frac{g_i}{2}A_i^\dagger(0) + \frac{1}{2}\int\frac{d\vartheta}{2\pi}K_i(\vartheta)A_{\bar{i}}^\dagger(-\vartheta)A_i^\dagger(\vartheta)\right\}|\Omega\rangle\,,$$ (5.8)

and the post-quench Hamiltonian is integrable ($\bar{i}$ denotes the antiparticle of $i$ and $g_i \neq 0$ only for self-conjugate particles $i = \bar{i}$). In the presence of single-particle overlaps $g_i$, the pair-state amplitudes $K_i(\vartheta)$ have a first order pole at the zero-momentum threshold [100]:

$$K_i(\vartheta) \sim \frac{-\iota g_i^2}{2\vartheta} + O(\vartheta^0)\,.$$ (5.9)

Such states are analoguous to the integrable boundary states introduced by Ghoshal and Zamolodchikov [101]; however, in contrast to states corresponding to boundary conditions the quench inital states have a finite norm. There are several reasons to call such a quench integrable:

- Factorised overlaps: the overlaps of multi-particle states are factorised into products of independent pair creation amplitudes (with a possible inclusion of single particle excitations with zero momentum if $g_i \neq 0$).

- These states preserve the odd conserved charges of the post-quench theory, just as the integrable boundary states do. Recently, the analogous notion of integrable initial state was extended to the lattice situation [82] and was found to include all known cases that had been solved exactly by a thermodynamic Bethe Ansatz built upon the quench action principle introduced in Ref. [102].

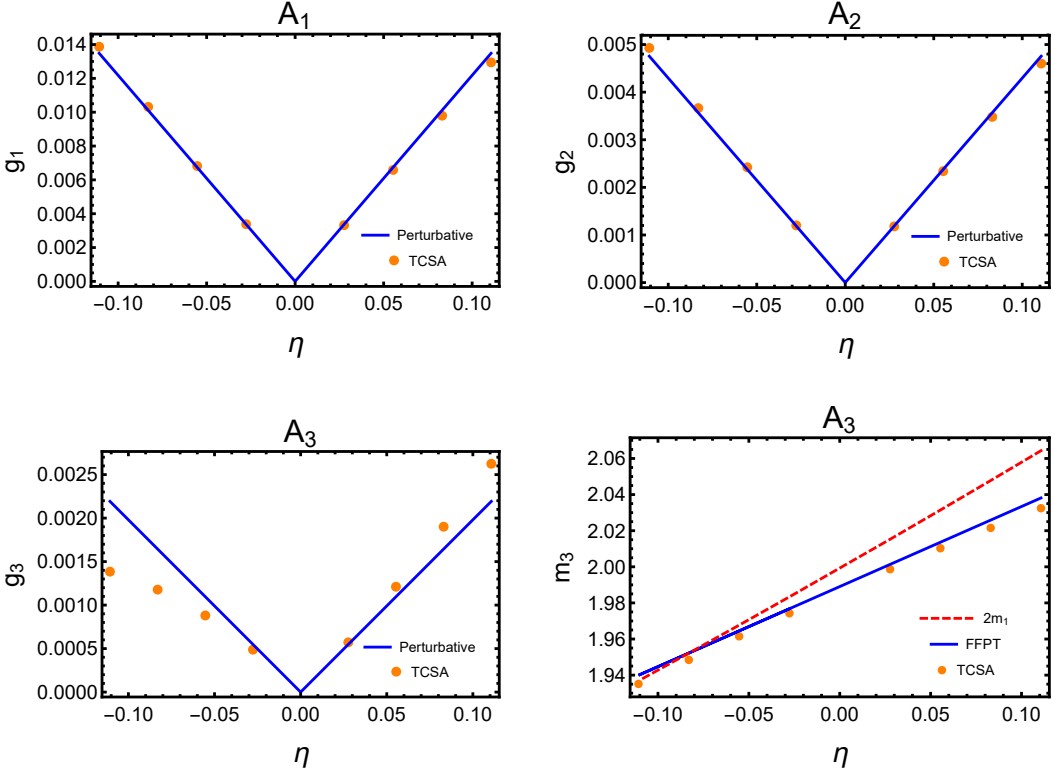

Figure 5.2: The first three panels show the comparison between TCSA and perturbative overlaps as functions of the quench strength $\eta$ for quenches away from the $E_8$ axis in volume $r = 40$. The fourth (bottom right) panel shows the dependence of the mass $m_3$ of the third particle as a function of $\eta$, compared to the form factor perturbation theory prediction (4.4) and the two-particle threshold $2m_1$ also calculated using (4.4).

• For an initial state of this form it is possible to apply the thermodynamic Bethe Ansatz to compute one-point functions in the equilibrium state [32, 103].

Apart from quenches in free field theory where the initial state has the form (5.8) due to the Bogoliubov transformation, it is known that quenches from a free massive field to the sinh–Gordon model are at least approximately well described by a generalised squeezed state [58, 60]. Using form factor methods the overlap function was also explicitly computed, and confirmed using TCSA after an analytic continuation to the sine–Gordon model [104]. In addition, for small post-quench density the semiclassical quasi-particle picture [14, 47, 48] is expected to hold together with an approximate factorisation of overlaps if the individual particle creation processes are weakly correlated.

The overlaps determine the so-called statistics of work, which gives the probability of the system to be found in an excited state of a given energy $W$ after the quench. In finite volume, the spectrum of the system is discrete and consequently the statistics of work $W$ performed in the quench is a sum of Dirac delta peaks,

$$P(W) = \sum_n |\langle \Psi(0)|n\rangle|^2 \, \delta(E_n - E_0 - W), \qquad (5.10)$$

where $|n\rangle$ is a post-quench eigenstate with energy $E_n$ and $E_0$ is the energy of the pre-quench vacuum state, $E_0 = \langle 0|H_0|0\rangle$.

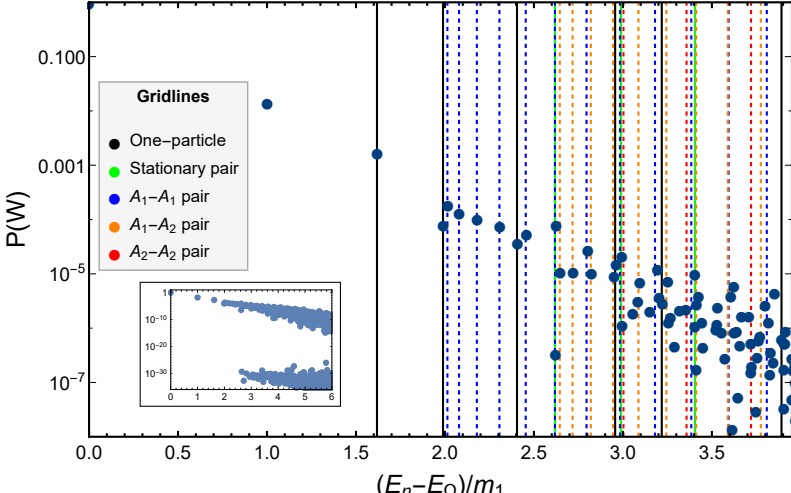

Figure 5.3: Statistics of work for a quench of size $\xi = 0.5$ along the $E_8$ axis with truncation level $N_{\text{cut}} = 18$ in volume $r = 40$. The first one-particle state sets the mass unit at $(E_1 - E_\Omega)/m_1 = 1$, while the energies of all other one-particle states are shown by continuous black gridlines. Green lines correspond to two-particle states with two zero-momentum particles of different species, while two-particle states involving moving particles are indicated by dashed lines, with colour code shown in the legend. One-particle states were identified by their mass, while for two-particle states we matched their energies with those predicted by the Bethe–Yang equation (C.6). The inset shows a larger range where two separate branches are observable.

An example for the $P(W)$ in a quench along the $E_8$ axis can be seen in Fig. 5.3 where the dots represent the weights of the Dirac deltas in (5.10). Note that the horizontal scale is shifted by $E_\Omega - E_0$. The inset plots a larger range which shows that there is a set of states that have negligible overlaps of the order of double precision arithmetic error. Since the respective eigenvectors are explicitly constructed in TCSA, we could directly verify that these states are odd under spatial reflection and so must have vanishing overlaps due to the quench dynamics preserving parity invariance.

We now turn to the measurement of the $K_{ij}(\vartheta)$ functions in Eq. (5.1) which give the amplitude for pair creation by the quench. Using the statistics of work data one can extract the absolute value of these functions for the two lightest species of particles $A_1$ and $A_2$. To obtain the infinite volume overlap functions, it is necessary to convert the finite volume results of Fig. 5.3 using the methods described in Appendix C, resulting in the data in Fig. 5.4. For these overlaps, values extracted at cutoff $N_{\text{cut}} = 17$ value were used.

In Fig. 5.4 we plot the $K_{ij}$ amplitudes as functions of the momentum $p$ which for a single particle of mass $m$ is related to the rapidity by $p = m \sinh(\vartheta)$. There are several interesting conclusions that can be drawn from the results depicted in the plot. First, the existence of non-zero overlaps for states $|A_1(-p)A_2(p)\rangle$ demonstrated in Fig. 5.4a implies that the quench is not integrable in the sense discussed at the beginning of this subsection. In fact, such states are not annihilated by the odd higher-spin charges due to the mass difference $m_1 \neq m_2$. This shows that integrability of the pre-quench and post-quench Hamiltonians does not imply integrability of the resulting quench, which was also argued on the basis of perturbation theory in Ref. [52].

Second, the data for the lightest pair overlap $K_{11}(p)$ plotted in Fig. 5.4b makes it possible to identify its asymptotic behaviour for large momenta which turns out to be exponentially decreasing. In the case of quenches in free field theories, the overlaps always decay as powers

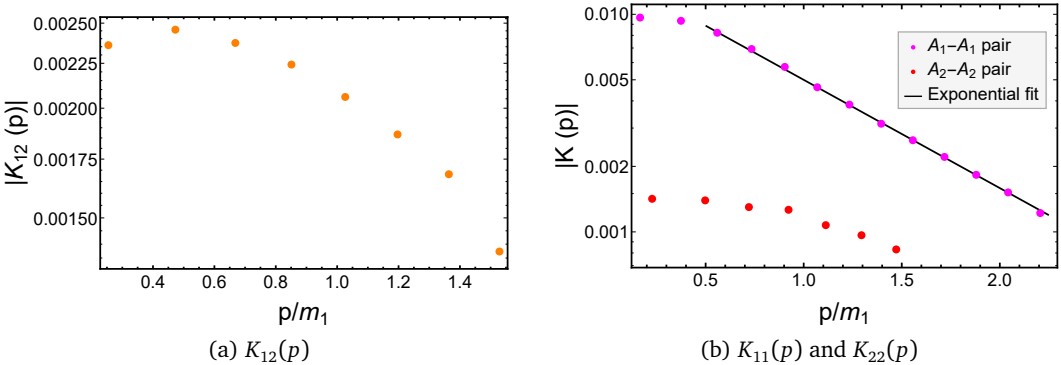

(a) $K_{12}(p)$          (b) $K_{11}(p)$ and $K_{22}(p)$

Figure 5.4: Overlap function of two-particle states (a) $|A_1(-p)A_2(p)\rangle$, (b) $|A_1(-p)A_1(p)\rangle$ and $|A_2(-p)A_2(p)\rangle$ as a function of momentum $p$ on a logarithmic plot, extracted in volume $r = 40$ after a quench along the $E_8$ axis for a quench of size $\xi = 0.5$.

of the momentum; in the sinh–Gordon and sine–Gordon quenches studied in the works [58, 60,104] this continued to hold since they were obtained as relevant perturbations of a massless free boson which determines their high-energy behaviour. At first sight this is also the case for the quench considered here since the fixed point conformal field theory is a massless Majorana fermion. However, the $E_8$ model has a particle basis $A_1, \ldots, A_8$ that is essentially different from the free fermions, and these particles do not become free in the ultraviolet limit. This is not a contradiction as in the limiting conformal field theory there is no unique notion of a particle basis, and taking the massless limit from different massive directions produces different results. In the language of massless $S$-matrices [105], the high energy limits taken along the massive Majorana and the $E_8$ directions to the free massless fixed point result in two very different scattering descriptions of the same conformal field theory.

## 6 Conclusions

In this work we investigated quenches starting from the Ising $E_8$ integrable quantum field theory using the Truncated Conformal Space Approach. Truncated Hamiltonian methods (in a slightly different version) have already been applied successfully to quenches starting from the other integrable line in the parameter space of the scaling Ising field theory which corresponds to a massive Majorana fermion [65]. It turns out that the $E_8$ model is even more accessible by the method due to its much better convergence properties, which allowed us to get very accurate numerical results for the time evolution of observables, the essential limitation being an upper time limit due to the finite volume of the system. This enables us to draw definite conclusions about the accuracy of two proposed form factor based expansions for the time evolution. Both form factor expansions are valid for suitably small post-quench particle density which is also a condition for the validity of truncated Hamiltonian approaches to quantum quenches (cf. Ref. [65]), and also of the so-called semiclassical approach [47, 48], so it is a general limitation in our understanding of quenches in strongly correlated quantum field theories.

The post-quench expansion approach [49, 50, 53] assumes that the post-quench Hamiltonian is integrable and applies an expansion in the post-quench asymptotic multi-particle basis. It presupposes a knowledge of the overlaps of the initial state with the post-quench eigenstates, which can be supplied by the Hamiltonian truncation as in our case, but there are also ana-

lytic results in limited cases [58, 60, 104]. Perturbation theory can also provide analytic results for the overlaps. Once the overlaps are specified (in our case from TCSA) then apart from a very brief initial transient the expansion was found to match the simulation data perfectly at least up to times $m_1 t = 30$ even for quenches corresponding to a change in the coupling of a magnitude comparable to its initial value.

We stress that there is a long time window of very precise agreement between theory and numerics, which means that the combination of the truncated Hamiltonian and the post-quench approach together provide a very precise numerical and (partially) also analytic understanding of the dynamics of small density quenches ending in a massive integrable QFT. Although the truncated version of the post-quench expansion used here is not valid for arbitrary long times, a resummation of infinitely many terms has been carried out [49, 50, 53] which turns it into a valid description to infinite times.

Let us now turn to the perturbative quench expansion approach [51, 52] which is the only general analytic approach to non-integrable quenches. We recall that this method is a perturbative expansion around the pre-quench theory which is assumed to be integrable. While its perturbative nature is compatible with the low density regime, it also puts an upper time limit on its validity.

As expected, we found excellent agreement between the perturbative predictions and the TCSA data for the smallest quenches we studied both for integrable and non-integrable quenches. However, we observed clear deviations for larger quenches even within the time window set by the perturbative framework. Comparison with numerics and with the post-quench expansion in the integrable case revealed that the main source of the discrepancy is a frequency mismatch: the perturbative approach uses the pre-quench masses whereas the time evolution is naturally governed by the post-quench energies. Similarly, the matrix elements of the operators appearing in the expansion are the pre-quench ones, while the post-quench quantities give a more accurate description in a truncated expansion.

The presence of the frequency shift explains why the first order perturbative result can become inaccurate much before the estimated upper limit $t^*$ (3.5) of the perturbative time window. Such a shift will show up in the higher orders starting with the second order as secular terms, i.e. as terms proportional to powers of $t$ which is easily seen by Taylor expanding the function $\cos[(\omega_0 + \lambda \omega_1)t]$ in $\lambda$. An example of such a deviation for times $t \ll t^*$ is the non-integrable quench discussed in Subsection 4.2.1.

The domain of validity of the perturbative approach can be extended by including higher orders. This would be an important development especially for two reasons. First, the perturbative approach can be applied to quenches with a non-integrable post-quench dynamics provided the pre-quench dynamics is integrable, and is therefore complementary to the post-quench expansion. Second, it does not require the overlaps as inputs; rather, it constructs them in the course of the perturbative expansion, and we demonstrated that the resulting overlaps are quite accurate when the quench size is suitably small.

The perturbative quench expansion is actually an application of form factor perturbation theory (introduced in Ref. [92]) to time evolution. We recall that post-quench frequencies were successfully reproduced in Ref. [65] using second order form factor perturbation theory developed for equilibrium mass corrections in Ref. [106]. Higher corrections would also include the perturbative reconstruction of the post-quench form factors in terms of the pre-quench theory.

However, as we showed above, higher order will include secular terms, so the time window of validity can only be extended by a resummation of the perturbation series taking into account infinitely many orders. Fortunately, this is not without precedent, it has been carried out in the post-quench formalism. On the basis of the results in Ref. [49, 50, 53] we expect that secular terms would account for the mass corrections also including additional frequency shifts due to the finite post-quench density, which can also have imaginary parts describing

relaxation. In fact, the comparison to truncated Hamiltonian calculations in Ref. [65] already revealed that the frequency shift and relaxation was missing from the first order results, albeit in that case the predicted oscillation amplitude was accurate. We propose that extending the perturbative approach to include higher order corrections improved by resummation of secular terms would bring it into agreement with the numerical results for larger quenches for an extended, possibly infinite time range, and therefore would provide an invaluable tool to examine quenches to non-integrable systems.

Besides the validity of the form factor expansions, we also examined two-particle overlaps obtained from TCSA and obtained some new insights. We have found an explicit example for a quench from an integrable to another integrable model where the overlaps do not have the particle pair state structure characteristic of the generalised squeezed states. Therefore such quenches cannot be considered integrable in the sense used in Ref. [82,104]. Arguments in favour of this scenario were put forward in Ref. [52]. In addition, we found that the two-particle overlap function decayed exponentially for large momenta, which contrasts with the case of sinh/sine–Gordon theory [58,60,104] or quenches on the free fermion axis $h = 0$ of the Ising model where the overlaps decay polynomially. We argued that this can be understood on the basis that the ultraviolet limit of the scattering theory is not free, in contrast to the sinh/sine-Gordon case, and so the high-energy limit of the overlaps need not behave the same way as for quenches in free field theories. It is an open issue whether the exponential decay is generally true when the ultraviolet limit is interacting in the relevant particle basis.

As a last remark we mention that the prethermalisation scenario can also be addressed perturbatively [25]. In this case the small parameter corresponds to the strength of integrability breaking in the post-quench Hamiltonian. We did not considered this issue here as the post-quench dynamics in Ising scaling field theory (cf. also [65]) has not been found to contain any regime where the prethermalisation scenario seems applicable. Identifying prethermalisation behaviour in quantum field theory quenches and comparing numerical results with the theoretical expectations are interesting issues to investigate in the future.

## Acknowledgments

**Funding information**  This research was partially supported by the BME-Nanotechnology FIKP grant of the Ministry of Human Capacities (BME FIKP-NAT), and also by the National Research Development and Innovation Office (NKFIH) under K-2016 grant no. 119204, OTKA grant no. SNN118028 and the Quantum Technology National Excellence Program (Project No. 2017-1.2.1- NKP-2017- 00001). M.K. was also supported by a Prémium postdoctoral grant of the Hungarian Academy of Sciences, while H.K. acknowledges support from the ÚNKP-17- 2-I New National Excellence Program of the Ministry of Human Capacities.

# A $E_8$ spectrum, scattering amplitudes and normalisation of form factors

The scaling Ising field theory with magnetic field has eight stable particles (denoted by $A_i$, $i = 1, 2, ..., 8$) organised according to the famous $E_8$ spectrum with the mass ratios [83]:

$$
\begin{aligned}
m_2 &= 2m_1 \cos\frac{\pi}{5} = (1.618033989...)m_1\,, \\
m_3 &= 2m_1 \cos\frac{\pi}{30} = (1.989043791...)m_1\,, \\
m_4 &= 2m_2 \cos\frac{7\pi}{30} = (2.404867172...)m_1\,, \\
m_5 &= 2m_2 \cos\frac{2\pi}{15} = (2.956295201...)m_1\,, \\
m_6 &= 2m_2 \cos\frac{\pi}{30} = (3.218340459...)m_1\,, \\
m_7 &= 4m_2 \cos\frac{\pi}{5}\cos\frac{7\pi}{30} = (3.891156823...)m_1\,, \\
m_8 &= 4m_2 \cos\frac{\pi}{5}\cos\frac{2\pi}{15} = (4.783386117...)m_1\,,
\end{aligned}
\tag{A.1}
$$

where all masses are expressed in terms of the mass gap $m_1$ (the mass of the lightest particle $A_1$) which is related to the coupling by Eq. (2.2). The Hilbert space is spanned by the asymptotic multi-particle states

$$
\left| A_{i_1}(\vartheta_1),\ldots,A_{i_n}(\vartheta_n) \right\rangle = A^\dagger{}_{i_1}(\vartheta_1)\ldots A^\dagger{}_{i_n}(\vartheta_n) \left| \Omega \right\rangle\,,
\tag{A.2}
$$

where $\vartheta_j$ are the rapidity parameters related to the energy and momentum by the relations $e = m\cosh\vartheta$ and $p = m\sinh\vartheta$ for a particle of mass $m$. We also quote the two-particle $S$-matrix amplitudes that are used in our calculations:

$$
\begin{aligned}
S_{11}(\vartheta) &= \left\{\frac{2}{3}\right\}\left\{\frac{2}{5}\right\}\left\{\frac{1}{15}\right\}\,, \\
S_{12}(\vartheta) &= \left\{\frac{4}{5}\right\}\left\{\frac{3}{5}\right\}\left\{\frac{7}{15}\right\}\left\{\frac{4}{15}\right\}\,, \\
S_{22}(\vartheta) &= \left\{\frac{4}{5}\right\}\left\{\frac{2}{3}\right\}\left\{\frac{7}{15}\right\}\left\{\frac{4}{15}\right\}\left\{\frac{1}{15}\right\}\left\{\frac{2}{5}\right\}^2\,, \\
\{x\} &= \frac{\sinh\vartheta + i\sin\pi x}{\sinh\vartheta - i\sin\pi x}\,,
\end{aligned}
\tag{A.3}
$$

where the argument is the rapidity difference of the incoming particles.

The form factors of a local operator $\Phi(x)$ are defined as the matrix elements

$$
F_{i_1\ldots i_n}(\vartheta_1,\ldots\vartheta_n) = \left\langle \Omega|\Phi(0)\left| A_{i_1}(\vartheta_1),\ldots,A_{i_n}(\vartheta_n)\right\rangle\right.\,.
\tag{A.4}
$$

One-particle form factors were obtained in Ref. [107, 108], while higher form factors were computed in Ref. [84] to which we refer the interested reader for details. In all these works, the operators were normalised to have unit expectation value. Since we normalise our operators according to their short-distance operator product expansion:

$$
\langle \Phi(x)\Phi(0) \rangle = \frac{1}{|x|^{4h_\Phi}} + \ldots\,,
\tag{A.5}
$$

these results must be rescaled using the exact vacuum expectation values

$$
F^\Phi = F^\Phi_{\langle\Phi\rangle=1}\langle\Phi\rangle\,.
\tag{A.6}
$$

The exact expectation values of the $\sigma$ and $\epsilon$ fields were obtained in Ref. [85]:

$$
\langle\sigma\rangle = (-1.06144\ldots)m_1^{1/8}\,, \quad \langle\epsilon\rangle = (0.454752\ldots)m_1\,.
\tag{A.7}
$$

Table B.1: Matrix size vs. cutoff

| cutoff level $N_{\text{cut}}$ | matrix size | cutoff level $N_{\text{cut}}$ | matrix size |
|---|---|---|---|
| 9 | 302 | 14 | 2734 |
| 10 | 487 | 15 | 4076 |
| 11 | 759 | 16 | 6029 |
| 12 | 1186 | 17 | 8774 |
| 13 | 1798 | 18 | 12820 |

# B    TCSA details

## B.1    Settings and conventions

The Truncated Conformal Space Approach was introduced by Yurov and Zamolodchikov in Ref. [66, 68]. The action in Eq. (2.1) in finite volume $R$ with periodic boundary conditions leads to the following dimensionless Hamiltonian:

$$H = H_{CFT} + H_\sigma + H_\epsilon = \frac{2\pi}{r}\left( L_0 + \bar{L}_0 - c/12 + a_1\kappa\frac{r^{2-2h_\sigma}}{(2\pi)^{1-2h_\sigma}}M_\sigma + a_2\frac{r^{2-2h_\epsilon}}{(2\pi)^{1-2h_\epsilon}}M_\epsilon \right), \quad \text{(B.1)}$$

$a_2 = 0$ corresponds to the system on the $E_8$ axis and the normalisation factor $\kappa = 0.06203236\ldots$ is chosen using (2.2) to ensure that the first particle mass is $m_1 = 1$ for $a_1 = 1$ provided the volume is measured using the units $r = m_1 R$. $L_0$ and $\bar{L}_0$ are the standard Virasoro generators, while the matrix elements of the perturbing operators are parameterised by $M_\sigma$ and $M_\epsilon$ and can be constructed using the conformal Ward identities.

The parameter $a_2$ can be written as

$$a_2 = -\frac{M}{2\pi m_1} = (-0.03613127\ldots)\eta \quad \text{(B.2)}$$

in terms of the dimensionless ratio $\eta = M/|h|^{8/15}$.

Due to translational invariance of the quenches considered here it is enough to keep only states with zero total momentum. The space is truncated to basis vectors below a certain conformal level, given by the cutoff parameter $N_{\text{cut}}$. The cutoff values used in our calculations and the corresponding matrix sizes are shown in Table B.1.

## B.2    Extrapolation in cutoff

The bare TCSA results depend on cutoff parameter $N_{\text{cut}}$ which can be (approximately) eliminated using renormalisation group methods [109–112]. We follow the procedures used previously in Ref. [65], to which we refer the interested reader for details; here we only give the details specific for the models considered in this paper.

The first level of correction is introducing running couplings in the Hamiltonian (B.1) using the leading order RG equations in Ref. [111] (cf. also [94]). It turns out that this correction is very small, but for complete consistency we nevertheless implemented it.

The expectation values have power law corrections in the cutoff whose exponents can be computed using the formalism in Ref. [113]. In the case of a quench along the $E_8$ axis the dependence on the cutoff level $N_{\text{cut}}$ is:

$$\langle\sigma(t)\rangle_{\text{TCSA}} = \langle\sigma(t)\rangle + A_\sigma(t)N_{\text{cut}}^{-7/4} + B_\sigma(t)N_{\text{cut}}^{-11/4} + \ldots, \quad \text{(B.3)}$$

and

$$\langle\epsilon(t)\rangle_{\text{TCSA}} = \langle\epsilon(t)\rangle + A_\epsilon(t)N_{\text{cut}}^{-1} + B_\epsilon(t)N_{\text{cut}}^{-2} + \ldots, \quad \text{(B.4)}$$

where the ellipsis denote further subleading corrections.

In principle, one has to perform extrapolation using both leading and subleading exponents of Eqs. (B.3) and (B.4). However, there are too few data points for a two-parameter fit to be stable. Conclusively, our results were achieved by extrapolation with leading exponents only. Extrapolated time-dependent curves were acquired by performing extrapolation at each sampling point in time. Illustrations can be seen in Figs B.1.,B.2. and B.3.

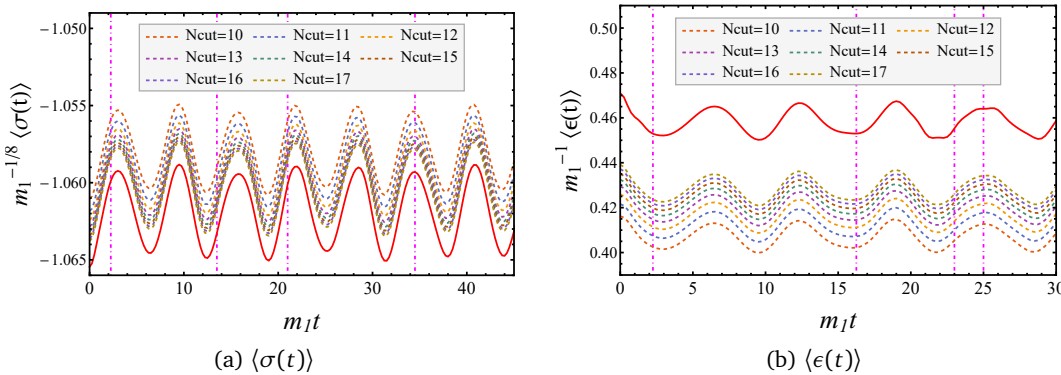

(a) $\langle \sigma(t) \rangle$        (b) $\langle \epsilon(t) \rangle$

Figure B.1: Time evolved operators at different cutoffs after a quench of size $\xi = 0.05$ in volume $r = 40$. The result of the extrapolation is shown in red. Extrapolation sections shown below are at time slices corresponding to dot-dashed gridlines. On the $E_8$ axis both operators can be calculated for very long times without the quality of extrapolation declining with time.

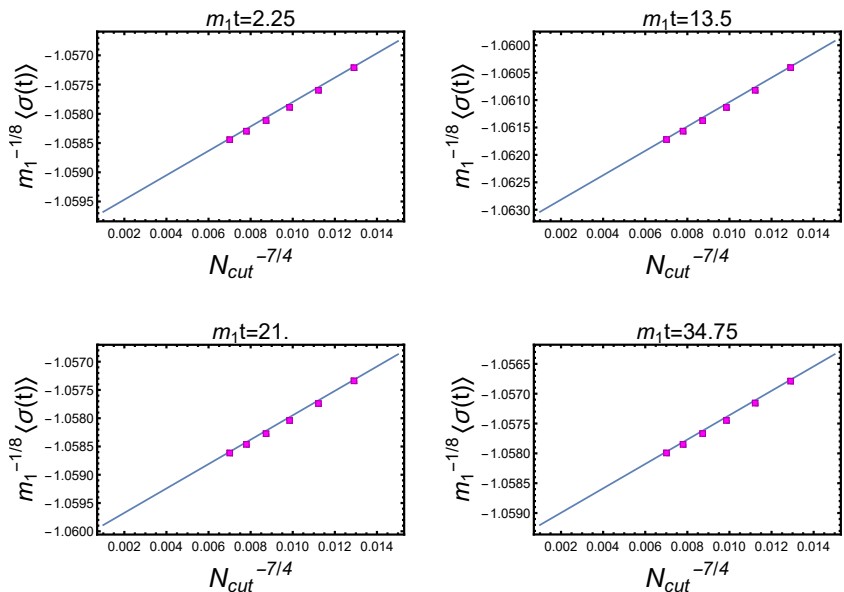

Figure B.2: Extrapolation in cutoff level at different time slices corresponding to the dot-dashed gridlines in Fig. B.1a. Points are in line even at longer times.

To explore limitations of the TCSA results we have to determine whether performing calculations at different volumes have any effect on the final results. Figs B.4a. and B.4b. illustrate volume-dependece of cutoff-extrapolated TCSA data. Although curves are not in perfect agreement, the main difference is a shift in the oscillation baseline. Despite this, we can still use

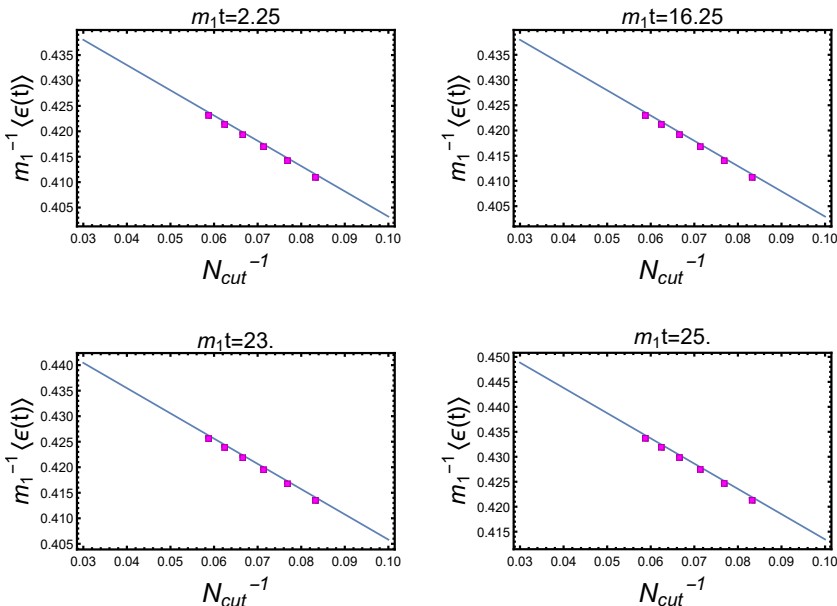

Figure B.3: Extrapolation in cutoff level at different time slices corresponding to the dot-dashed gridlines in Fig. B.1b. The quality of fit is as good as earlier.

TCSA to make comparisons between perturbative predictions since diagonal ensemble average is also volume-dependent.

A similar procedure can be applied when we perform quenches to regions of the $h - m$ plane where integrability is broken (i.e. $\eta \neq 0$ quenches). There is an important difference, however: now the post-quench Hamiltonian contains an $\epsilon$ perturbation, so different powers are obtained for the cutoff level dependence:

$$\langle \sigma(t) \rangle_{\text{TCSA}} = \langle \sigma(t) \rangle + A_\sigma(t) N_{\text{cut}}^{-1} + B_\sigma(t) N_{\text{cut}}^{-7/4} + O(N^{-2}), \tag{B.5}$$

and

$$\langle \epsilon(t) \rangle_{\text{TCSA}} = \langle \epsilon(t) \rangle + A_\epsilon(t) \ln N_{\text{cut}} + B_\epsilon(t) N_{\text{cut}}^{-1} + C_\epsilon(t) N_{\text{cut}}^{-2} + O(N^{-3}). \tag{B.6}$$

Eq. (B.6) makes it clear that the one-point function of $\epsilon$ for $M \neq 0$ is logarithmically divergent, which complicates the evaluation of the expectation value $\langle \epsilon(t) \rangle$ since it diverges

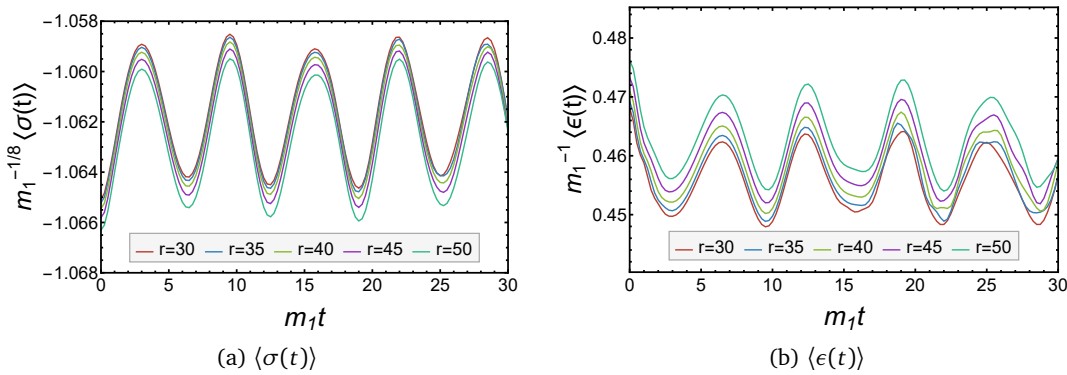

Figure B.4: Extrapolated curves at different volumes for $\xi = 0.05$ are very similar except for a constant shift. The volume dependence is more observable in the fact the the curves tend to deviate from the others after $t = R/2$.

with the cutoff. However, the divergences can be absorbed by the shift $C_\epsilon$ in (3.3) and the diagonal average in (3.6), while the fluctuations are well-defined.

To obtain a stable result from our TCSA data we used the following procedure. First we subtracted the divergent constant term (which is straightforward, as its value is well-defined as a function of cutoff and volume) by taking time average over some finite number of oscillations, neglecting the transient effects at the beginning of each curve. We then extrapolated these subtracted curves in the cutoff using the $N_{\text{cut}}^{-1}$ dependence. In principle one should also perform an extrapolation in volume but that proved to be unnecessary as there was no observable volume dependence left. This procedure defines the regularised value of $\langle \epsilon \rangle_D$ to be zero, while $C_\epsilon$ can be obtained by fitting the beginning of the curve (3.3) to the regularised value of $\langle \epsilon(0) \rangle$.

For the operator $\sigma$ we generally used the leading exponent, except in the case of smallest quenches $|\eta| = 0.0044$ where due to the small value of $M$ this leading term has a very small coefficient $A_\sigma$ so we performed extrapolation using $N^{-7/4}$ instead. Illustration of this phenomenon is given in Fig. B.5.

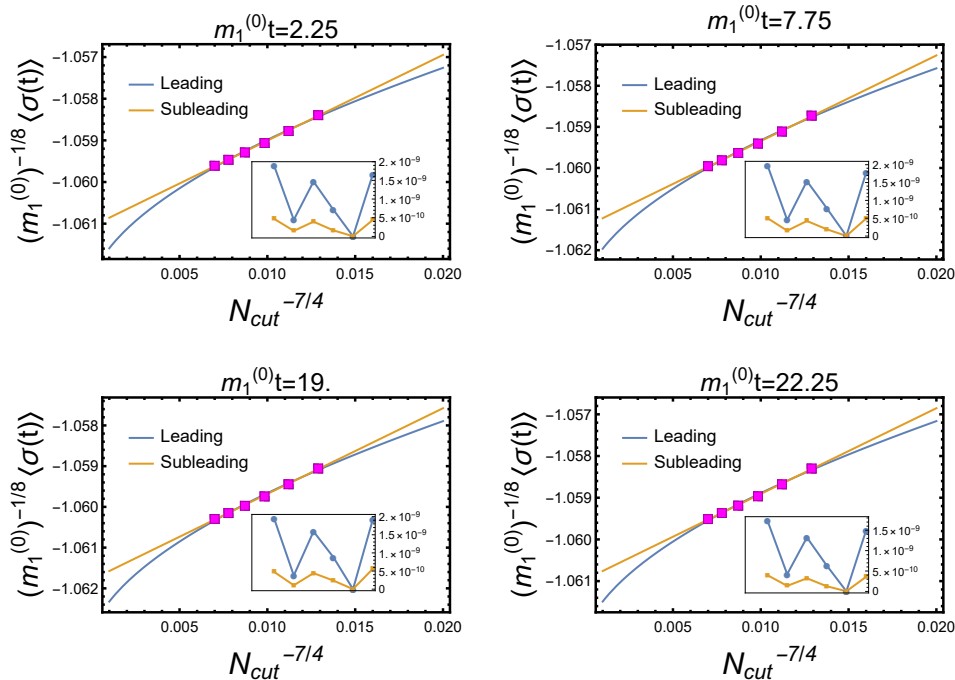

Figure B.5: Comparison of extrapolation coefficients for an $\eta = -0.0044$ quench, $r = 40$. Insets show fit residuals squared at each point. Apparently, subleading exponents fit our data better.

Volume-dependence is a bit stronger than earlier, especially that of $\langle \epsilon(t) \rangle$. This can also be seen in Fig. 4.5b; further illustration is provided in Fig. B.6.

## C  Overlaps in finite volume

TCSA computations give us overlaps in a finite volume $R$ with periodic boundary conditions, which are related to the infinite volume amplitudes $\tilde{K}$ in Eq. (5.1) via the theory of finite size dependence of boundary state amplitudes worked out in Ref. [99] (see also Ref. [104] for some more details). Here we only give the formulae for the one and two-particle cases that

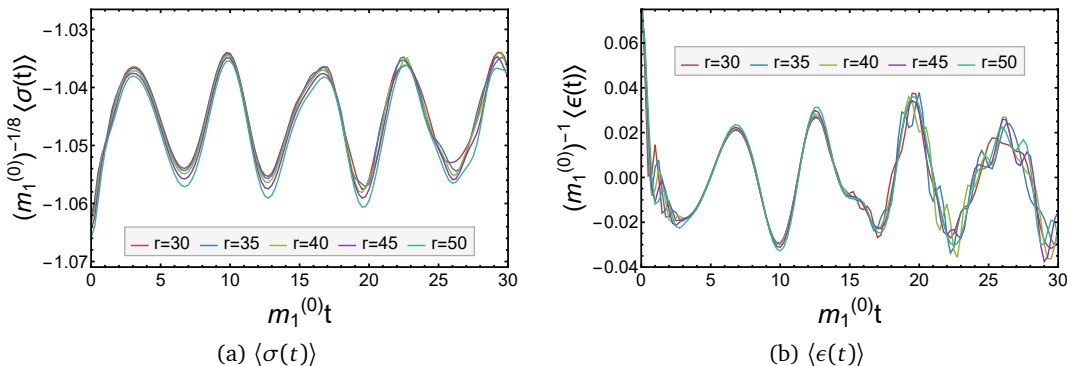

(a) $\langle \sigma(t) \rangle$                                                    (b) $\langle \epsilon(t) \rangle$

Figure B.6: Extrapolated curves at different volumes for a quench of size $\eta = -0.125$. Volume-dependence is clearly observable thus we concluded to consider TCSA results reliable until $t = R/2$.

are used in the main text.

For a one-particle state one can directly use the result derived in Ref. [99]

$$\frac{g_i}{2} \equiv \langle \Psi(0)|A_i(0)\rangle = \frac{1}{\sqrt{m_i R}} \langle \Psi(0)|A_i(0)\rangle_R \,, \tag{C.1}$$

where the subscript $R$ denotes matrix elements in finite volume.

For a two-particle state composed of $A_k$ and $A_l$, the relevant term in the initial state (5.1) can be written as

$$\int \frac{d\vartheta_1}{2\pi} \int \frac{d\vartheta_2}{2\pi} \tilde{K}_{kl}(\vartheta_1, \vartheta_2) |A_k(\vartheta_1)A_l(\vartheta_2)\rangle \,. \tag{C.2}$$

Due to translational invariance,

$$\tilde{K}_{kl}(\vartheta_1, \vartheta_2) = \bar{K}_{kl}(\vartheta_1, \vartheta_2)\delta(m_k \sinh\vartheta_1 + m_l \sinh\vartheta_2)\,, \tag{C.3}$$

and one rapidity integration can be eliminated, however, it is necessary to follow carefully which one it is. Performing the integral over $\vartheta_2$ gives

$$\int \frac{d\vartheta_1}{2\pi} K_{kl}^{(l)}(\vartheta_1) |A_k(\vartheta_1)A_l(\vartheta_2)\rangle \,, \tag{C.4}$$

where now $\vartheta_2 = -\text{arcsinh}(m_1 \sinh\vartheta_1/m_2)$ and

$$K_{kl}^{(l)}(\vartheta_1) = \frac{\bar{K}_{kl}(\vartheta_1, \vartheta_2)}{m_l \cosh\vartheta_2}\,. \tag{C.5}$$

In finite volume, the possible values of the rapidities are constrained by the quantization conditions

$$Q_1 = m_k R \sinh\vartheta_1 + \delta_{kl}(\vartheta_1 - \vartheta_2) = 2\pi I_1\,,$$
$$Q_2 = m_l R \sinh\vartheta_2 + \delta_{kl}(\vartheta_2 - \vartheta_1) = 2\pi I_2 \tag{C.6}$$

with $I_{1,2}$ momentum quantum numbers, where $\delta_{kl} = -i \log S_{kl}$ is the phase shift function. Solving this equation makes it possible to predict the energies of two-particle levels in finite volume as $m_k \cosh\vartheta_1 + m_l \cosh\vartheta_2$ relative to the vacuum (up to corrections exponentially small in large volume), leading to the state identifications in Fig. 5.3.

Following the reasoning outlined in Ref. [99] the relation between the finite and infinite volume two-particle overlaps have the form

$$K_{kl}^{(l)}(\vartheta_1) = \frac{\bar\rho_1(\vartheta_1, \vartheta_2)}{\sqrt{\rho_{12}(\vartheta_1, \vartheta_2)}} \, \langle 0|A_k(\vartheta_1) A_l(\vartheta_2)\rangle_R \,, \tag{C.7}$$

where

$$\rho_{12}(\vartheta_1, \vartheta_2) = \det\left\{\frac{\partial Q_a(\vartheta_1, \vartheta_2)}{\partial \vartheta_b}\right\}_{a,b=1,2}$$
$$= m_k R \cosh\vartheta_1 \, m_l R \cosh\vartheta_2 + (m_k R \cosh\vartheta_1 + m_l R \cosh\vartheta_2)\varphi_{kl}(\vartheta_1 - \vartheta_2),$$

where

$$\varphi_{kl}(\vartheta) = \frac{\partial \delta_{kl}(\vartheta)}{\partial \vartheta} \tag{C.8}$$

and

$$\bar\rho_1(\vartheta_1, \vartheta_2) = \left.\frac{\partial Q_1}{\partial \vartheta_1}\right|_{m_k \sinh\vartheta_1 + m_l \sinh\vartheta_2 = 0} = m_k R \cosh\vartheta_1 + \left(1 + \frac{m_k R \cosh\vartheta_1}{m_l R \cosh\vartheta_2}\right)\varphi_{kl}(\vartheta_1 - \vartheta_2) \tag{C.9}$$

with $\vartheta_2 = -\mathrm{arcsinh}(m_1 \sinh\vartheta_1/m_2)$. Note that for the case of two identical particles ($k = l$) this result reduces to the one derived in Ref. [99], and in this case it is not necessary to track which rapidity integral is eliminated using momentum conservation. The overlap functions shown in Fig. 5.4 are

$$K_{11} = K_{11}^{(1)}, \quad K_{22} = K_{22}^{(2)}, \quad K_{12} = K_{12}^{(2)}. \tag{C.10}$$

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
