# Peer review of "Quench dynamics of the Ising field theory in a magnetic field"

_SciPost Physics, doi:SciPost Phys. 5, 027 (2018)_

## Round 3 · Referee Report · Dirk Schuricht (Referee 1) · 2018-4-25

Strengths

-Timely topic.
-Very careful analysis.
-Detailed comparison to previous approaches.

Weaknesses

-Minor improvements of the presentation possible (see below).

Report

The authors use the TCSA method to study the time evolution of the one-dimensional Ising field theory in a transverse and longitudinal magnetic field. They analyse quenches both in the integrable system but also away from integrability. After the quench they consider the Loschmidt echo as well as the magnetisation operators in both directions. In the integrable quenches they compare the TCSA results to a perturbative approach put forward by Delfino and the “post-quench” approach developed in Refs. 48,49,52. For non-integrable quenches only comparison to the perturbative approach is possible. They analyse the results in great detail, also considering the convergence of the TCSA approach carefully. The also study the amplitudes of certain single and two-particle contributions to the initial state of the post-quench time evolution. In particular, they find that even for quenches along an integrable line the initial state is not of the often considered squeezed state form.

I think the paper adds a lot of interesting results to the study of quenches in one-dimensional integrable and non-integrable systems. In particular, I value the comparison to complementary analytical approaches and the analysis of the initial state properties. Thus I conclude that the manuscript should be published in SciPost Physics once the remarks below have been addressed.

Requested changes

I have a few remarks that I ask the authors to consider:
-In addition to Refs. 24-27 also PRB 89,165104 (2014) seems related.
-In Ref. 53 some informations (published etc) are missing.
-When discussing the use of truncation methods for quantum quenches, I am wondering whether Ref. 23 also used such methods.
-I don’t understand the statement “Note that this does not involve any Yang-Baxter type compatibility relation between the squeezed state’s two-particle amplitude and the post-quench S-matrix.”, since in my understanding the very definition of a squeezed state requires precisely such a compatibility condition.
-When discussing the two types of quenches in Sec. 2, maybe a sketch of the parameter space with an indication of these quenches would be helpful to the reader.
-After (2.5), how is m_1 defined in the second type of quenches?
-In (3.6) I suppose that the last sum is only over indices i\neq j, since the summation over i=j is already captured in the first sum? The same remark applies to (3.13).
-After (3.13) Ref. 101 appears, ie, the order of references seems to be mixed up.
-When discussing Figs. 3.2 and 3.3, is it possible to give error bars or maybe add a remark on the error of the TCSA data in the captions?
-Is my understanding correct that the deviations in Figs. 3.2a and 3.3a are due to the TCSA error, ie, there is no problem with the analytical approaches?
-From the discussion of Fig. 3.3 it seems that the main error of the TCSA data comes from the cutoff extrapolation. Is this the case?
-In the caption of Fig. 3.2 the units of the spin expectation value are discussed. Maybe it would be better to include the units directly at the axis of the plot.
-When comparing the different approaches the authors refer to the low-density regime. Is it possible to determine the density of excitations from the TCSA, maybe from the data of Sec. 5.2? If so, I would find it very useful to add such a discussion in Sec. 3.3 or 5.2.
-In Fig. 3.6, is there any understanding why the the Loschmidt echo does not possess a peak at m_2-m_1? Also in that figure, the data above \omega=2 becomes rather erratic. The authors should comment on this observation and give an explanation if possible.
-Concerning the observed shifts in the one-particle peaks, is it possible to compare to Ref. 52, at least at a qualitative level?
-In the discussion of Sec. 3.4, in which sense is the projection onto the initial state non-local? Just in a spatial sense or also in the sense of non-local operators, form factors and so on?
-In the figures in Sec. 4, are the units given by the 1st order result for the mass m_1 given in (4.4)? The reference to the units of Fig. 3.6 is a bit confusing, since there the pure mass appears.
-In Fig. 4.7b, is it possible to relate the quasi-particle peaks to the meson masses? Is there any understanding why the peaks are wider that in the paramagnetic case?
-Maybe it would be useful to include the analytic results for the meson masses obtained in Refs. 87-89 for completeness. Also this would allow a direct comparison with the TCSA results.
-I didn’t find a mentioning of how the creation operators in (5.2) are defined and which algebra they satisfy.
-I suppose the state i in (5.4) should read A_i(0)?
-The labels (a)-(d) in Fig. 5.2 seems oddly placed.
-In (5.7) the labels i appear too far to the right. Also, does the integrable initial state (5.2) require any relation between the amplitudes g_i and K_i?
-Above (B.3) the term “power-like” should presumably read “power-law”.

  • validity: top
  • significance: high
  • originality: high
  • clarity: high
  • formatting: excellent
  • grammar: excellent

Author:  Márton Kormos  on 2018-08-04  [id 305]

(in reply to Report 1 by Dirk Schuricht on 2018-04-25)

We thank the Referee for his report and for his opinion that our manuscript should be published in SciPost.
Below we reply to his comments.

-In addition to Refs. 24-27 also PRB 89,165104 (2014) seems related.

Indeed, we added it to the list of references.

-In Ref. 53 some informations (published etc) are missing.

We have completed the reference (now Ref. 54).

-When discussing the use of truncation methods for quantum quenches, I
am wondering whether Ref. 23 also used such methods.

We completely agree, we have added a sentence to the Introduction
mentioning this work.

-I don’t understand the statement “Note that this does not involve any
Yang-Baxter type compatibility relation between the squeezed state’s
two-particle amplitude and the post-quench S-matrix.”, since in my
understanding the very definition of a squeezed state requires precisely
such a compatibility condition.

We have removed this remark because it was a bit misleading and
unnecessarily complicated the discussion. What we meant was that the
pair creation amplitude is not derived from a reflection factor that
satisfies the boundary Yang-Baxter equations. Relations required by
consistency, e.g. $K(\theta)=S(2\theta)K(-\theta)$ are of course satisfied.

-When discussing the two types of quenches in Sec. 2, maybe a sketch of
the parameter space with an indication of these quenches would be
helpful to the reader.

We have added a new figure, Fig. 2.1, that illustrates the quenches in
the parameter space.

-After (2.5), how is m_1 defined in the second type of quenches?

Thanks to the Referee, we realized that we hadn't been clear about our
units. For Type I quenches we use the post-quench mass $m_1$ while for
Type II quenches we use the pre-quench mass gap $m^{(0)}_1$ to form
dimensionless quantities. We now explain this below Eq. (2.5).

-In (3.6) I suppose that the last sum is only over indices i\neq j,
since the summation over i=j is already captured in the first sum? The
same remark applies to (3.13).

It has been corrected.

-After (3.13) Ref. 101 appears, ie, the order of references seems to be
mixed up.

This has been corrected.

-When discussing Figs. 3.2 and 3.3, is it possible to give error bars or
maybe add a remark on the error of the TCSA data in the captions?
-Is my understanding correct that the deviations in Figs. 3.2a and 3.3a
are due to the TCSA error, ie, there is no problem with the analytical
approaches?
-From the discussion of Fig. 3.3 it seems that the main error of the
TCSA data comes from the cutoff extrapolation. Is this the case?

We added the following sentences to the discussion of Figs. 3.2 and 3.3:
"In principle, the TCSA data have a residual error resulting from cutoff
extrapolation, but cutoff-dependence is mainly restricted to a shift in
the time-independent baseline of oscillations. As for the oscillations,
their error is comparable to the linewidth of Fig 3.2. Since the
baseline is adjusted for comparison (see later), we decided to omit
error bars in our plots."

We hope that this clarifies the issues regarding TCSA errors and answers
the above questions.

-In the caption of Fig. 3.2 the units of the spin expectation value are
discussed. Maybe it would be better to include the units directly at the
axis of the plot.

We now show explicitly the units on both axes in all of our figures.

-When comparing the different approaches the authors refer to the
low-density regime. Is it possible to determine the density of
excitations from the TCSA, maybe from the data of Sec. 5.2? If so, I
would find it very useful to add such a discussion in Sec. 3.3 or 5.2.

The data in Sec. 5.2 are rather incomplete in this regard. In
particular, we do not know the overlap functions K for small and large
values of the rapidity, and there is no information of overlaps
containing particles higher than the first two.
In addition, we expect a pole in the origin according to a recent work
of ours (our Ref. 100), but here we cannot access small enough
rapidities to see it. The overlaps shown in Fig. 5.4 are numerically
small, which is consistent with small density, as well as the validity
of both the form factor expensions (which depends on that), and also the
good convergence of the TCSA all indicate that the density is small
enough. We cannot really say more than the sentence in 3.3.1: "The
excellent agreement also implies that this quench is still in the low
density regime."

-In Fig. 3.6, is there any understanding why the the Loschmidt echo does
not possess a peak at m_2-m_1? Also in that figure, the data above
\omega=2 becomes rather erratic. The authors should comment on this
observation and give an explanation if possible.

The absence of the mass difference peaks in the Loschmidt echo can be
understood based on its spectral expansion and the smallness of the
overlaps $g_i$. The "erratic" behaviour is due to one-particle peaks
mixing with the multi-particle continuum which itself is a sum of
isolated peaks in finite volume. We added a discussion of both
observations at the end of Sec. 3.

-Concerning the observed shifts in the one-particle peaks, is it
possible to compare to Ref. 52, at least at a qualitative level?

As it was shown in the recent preprint [100], the final expressions of
Ref. [53] are not consistent with the pole of the $K(\theta)$ amplitudes
at the origin. The new calculation correcting for this could not provide
a definite prediction for the frequency shifts because this would
require at least a conjecture for the resummation of infinitely many
terms in a form factor expansion, cf. Ref [100].

-In the discussion of Sec. 3.4, in which sense is the projection onto
the initial state non-local? Just in a spatial sense or also in the
sense of non-local operators, form factors and so on?

This was a confusing remark which has been removed.

-In the figures in Sec. 4, are the units given by the 1st order result
for the mass m_1 given in (4.4)? The reference to the units of Fig. 3.6
is a bit confusing, since there the pure mass appears.

As discussed after (2.5), we measure everything in units of the
pre-quench mass for Type II quenches. This is now also explicitly
indicated in the axis labels.

-In Fig. 4.7b, is it possible to relate the quasi-particle peaks to the
meson masses? Is there any understanding why the peaks are wider that in
the paramagnetic case?

The vertical lines indicate the meson masses as measured in TCSA. The
small deviations are due to the presence of a finite density background.
The peaks seem wider than in the paramagnetic case only because of the
different range of plotted frequencies.

-Maybe it would be useful to include the analytic results for the meson
masses obtained in Refs. 87-89 for completeness. Also this would allow a
direct comparison with the TCSA results.

As it was shown in Ref. [94], the TCSA meson masses agree with
theoretical predictions to a high precision, so displaying the analytic
results would not result in any visible change. We added a sentence
regarding this at the end of Sec. 4.

-I didn’t find a mentioning of how the creation operators in (5.2) are
defined and which algebra they satisfy.

We have added Eqs. (5.2) and (5.3) for completeness.

-I suppose the state i in (5.4) should read A_i(0)?

Yes, it has been corrected.

-The labels (a)-(d) in Fig. 5.2 seems oddly placed.

This was due to the relatively long tick labels. We have removed the
subfigure labels.

-In (5.7) the labels i appear too far to the right. Also, does the
integrable initial state (5.2) require any relation between the
amplitudes g_i and K_i?

Indeed it does. We have added a sentence about the findings of the
recent preprint [100], namely that the presence of single-particle
overlaps $g_i$ implies that the pair-state amplitudes $K_i(\theta)$
have a first order pole at zero rapidity.

-Above (B.3) the term “power-like” should presumably read “power-law”.

Corrected.

---

## Round 3 · Referee Report · Anonymous (Referee 2) · 2018-6-26

Strengths

Highly topical
Well written and well organized

Weaknesses

None really -- I might ask for a better discussion of the unitary perturbation theory presented in Ref. 25 as an analytical technique for the computation of quench dynamics.

Report

In the manuscript "Quench dynamics of the Ising field theory in a magnetic field" by K. Hodsagi et al, the authors consider quantum quenches involving the integrable E8 model. The authors employ a combination of the truncated conformal spectrum approach and analytical approaches. The analytic
approaches include one suitable for when the post-quench Hamiltonian is integrable and for when the
matrix elements of post-quench observables can be computed and one where the pre-quench Hamiltonian
is integrable. Careful comparisons are made between the analytics and numerics
in an effort to ascertain the region of validity of both.

Overall I think this work has been well executed. After the changes/comments below are considered, I would
recommend that the manuscript be accepted.

Requested changes

i) The initial state in Eqn. 5.1 is written as the vacuum state. A better notation might be $|i\rangle$.

ii) In Fig. 5.1 it is a bit surprising that the overlaps $g_3$ and $g_4$ agree better with perturbation theory. I would think
that because the excitations are higher in energy their agreement with
perturbation theory would be worse. Could the authors comment on this.

iii) The first time (of which I am immediately aware) where the finite volume regularization was employed in comparing TCSA
data to form factor bootstrap computations for 1-pt functions was Phys. Rev. E 63, 016103.
This is relevant for the citation in the discussion surrounding
Eqn. 5.4.

iv) I think it would be worthwhile in the discussion of analytical approaches for studying quantum quenches to make contact
with the contents of Phys. Rev. B 84 054304 - Ref. 25 of the manuscript.
In Ref. 25 the authors use a unitary perturbation theory to study quantum quenches.
The approach is akin to one discussed in this manuscript where the matrix elements of the pre-quench Hamiltonian are required input (i.e. the Delfino-Viti methodology). However it seems distinct. A possible impediment to a direct comparison between Delfino-Viti and this PRB is that the authors there discuss the simplified situation of computing observables which commute with the pre-quench conserved quantities. However I think the authors should at least make better contact with this work as it predates the Delfino-Viti paper by a considerable period.

  • validity: top
  • significance: high
  • originality: good
  • clarity: top
  • formatting: excellent
  • grammar: excellent

Author:  Márton Kormos  on 2018-08-04  [id 304]

(in reply to Report 2 on 2018-06-26)

We thank the Referee for their report and for their recommendation for publication of our manuscript.
In the following we address the Referee's comments and requests.

i) The initial state in Eqn. 5.1 is written as the vacuum state. A
better notation might be |i⟩.

We agree with the Referee in that the original notation was slightly
misleading so we have changed it to $|\Psi_0\rangle$.

ii) In Fig. 5.1 it is a bit surprising that the overlaps g3 and g4 agree
better with perturbation theory. I would think that because the
excitations are higher in energy their agreement with
perturbation theory would be worse. Could the authors comment on this.

We do not have a satisfactory explanation for this at the moment. It is
possible that higher order corrections happen to be smaller for g3 and
g4 than for g2. We are currently working on the perturbative calculation
of the overlaps beyond the leading order, which will shed light on this
issue.

iii) The first time (of which I am immediately aware) where the finite
volume regularization was employed in comparing TCSA data to form factor
bootstrap computations for 1-pt functions was Phys. Rev. E 63, 016103.
This is relevant for the citation in the discussion surrounding Eqn. 5.4.

We thank the Referee for reminding us of this reference which we now
cite in relation with Eq. (5.5).

iv) I think it would be worthwhile in the discussion of analytical
approaches for studying quantum quenches to make contact
with the contents of Phys. Rev. B 84 054304 - Ref. 25 of the manuscript.
In Ref. 25 the authors use a unitary perturbation theory to study
quantum quenches.
The approach is akin to one discussed in this manuscript where the
matrix elements of the pre-quench Hamiltonian are required input (i.e.
the Delfino-Viti methodology). However it seems distinct. A possible
impediment to a direct comparison between Delfino-Viti and this PRB is
that the authors there discuss the simplified situation of computing
observables which commute with the pre-quench conserved quantities.
However I think the authors should at least make better contact with
this work as it predates the Delfino-Viti paper by a considerable period.

In fact, we have had the aim of testing prethermalisation scenarios
using TCSA. We were quite frustrated by the failure of identifying such
a regime in our numerics so far; instead, we found dynamical confinement
(clearly non-perturbative, cf. Ref. 88) or persistent/slowly decaying
oscillations (Ref. 65). The restrictions on the observables in Ref. 25
is also an impediment, since the ones we can easily access do not
satisfy that. Nevertheless, we think the whole issue is important enough
to deserve a discussion as an open issue and we have put in a paragraph
at the end of the Conclusions to explain the relation with this work.

---

## Round 4 · Author Response

Dear Editor,

We resubmit our manuscript after addressing all questions, comments and requests of the Referees. We found the reports very constructive and useful.

As both Referees suggested the publication of our manuscript, we hope that revised version can appear in SciPost Physics.

Yours sincerely,
The authors

---

## Round 4 · List of Changes

In our point-by-point replies to the reports we indicate all the changes.

---

## Editorial Decision

published